# SymSpectra: Symmetric Information Bottleneck Framework for Molecular Structure Recognition under Imbalanced Settings

## Abstract

Identifying molecular structures from spectral data is essential for early-stage chemical analysis, yet it remains a difficult task due to the imbalance in functional group distributions. Current methods often overfit to prevalent groups while neglecting underrepresented ones, failing to capture key dependencies between functional groups. This highlights the need for a unified approach that addresses both data imbalance and structural constraints. In this work, we present **SymSpectra**, a **Sym**metric Conditional Information Bottleneck (SCIB) framework designed to seamlessly integrate multi-modal **Spectra** features. Our model employs the SCIB framework to fuse multi-modal spectroscopic data into a unified representation, effectively preserving discriminative signals while mitigating redundancy. To enhance robustness against data imbalance, we incorporate conditional mutual information into the training objective, increasing the model's sensitivity to rare functional groups and challenging molecular cases. Additionally, a specialized module captures the dependencies among functional groups, improving both prediction accuracy and chemically meaningful interpretability. Experiments on multimodal spectral datasets demonstrate that SymSpectra significantly outperforms state-of-the-art methods, achieving an F1-score of 0.970 in substructure classification. More importantly, SymSpectra consistently outperforms baselines under various imbalanced scenarios, exhibiting superior robustness and generalizability, which may help advance the automation of chemical discovery. Our code can be found at https://anonymous.4open.science.

## 1 Introduction

The rapid advancement of artificial intelligence has revolutionized the interpretation of complex chemical data, facilitating tasks such as molecular property prediction and reaction planning Venkatasubramanian & Mann (2022); De Almeida et al. (2019); Brown et al. (2020). Central to these applications is the ability to extract structural insights from the low-dimensional spectral information. However, different spectroscopic techniques provide complementary insights. For example, infrared (IR) spectroscopy characterizes molecular vibrational modes Baiz et al. (2020), $^1$H-NMR elucidates the local environments of hydrogen atoms Yesinowski & Eckert (1987), and $^{13}$C-NMR captures the architecture of carbon frameworks Buddrus & Bauer (1987). Effectively integrating these diverse modalities allows AI models to leverage their synergistic information for accurate and efficient molecular identification Dale & Halgren (2001); Albers et al. (2022).

However, these data-driven approaches inevitably suffer from the challenge of imbalanced data distributions Pourkamali-Anaraki & Hariri-Ardebili (2021); Zhou et al. (2021). As illustrated in Figure 1 (a), functional groups in molecular datasets exhibit a highly skewed distribution: a few common groups dominate the majority of samples, while many groups—such as Azo compounds, Enol, and Phosphine—occur infrequently. This imbalance adversely affects model performance on rare functional groups, which may hold significant chemical importance. Figure 1 (b) demonstrates that classification accuracy for high-frequency groups far surpasses that of low-frequency ones, with an F1-score gap of 38%. Such disparity undermines the reliability of model predictions, especially when applied to novel molecules containing rare functional groups, thereby limiting practical utility in tasks like molecular discovery and design.

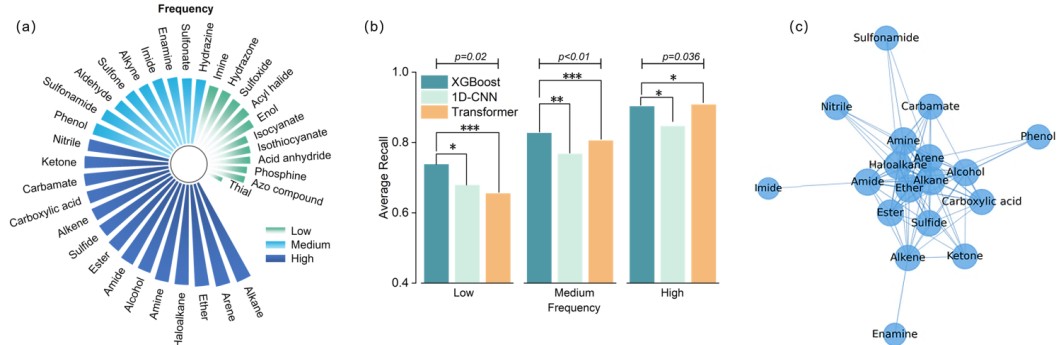

Figure 1: (a) The distribution of functional groups derived from the Alberts et al. dataset, categorized into three tiers to illustrate the inherent class imbalance. (b) The performance of three baseline models across these tiers. (c) The dependencies among functional groups, derived from co-occurrence statistics within the dataset. Edges connect groups with co-occurrence frequencies above a threshold, and node proximity indicates the strength of their association.

Furthermore, data imbalance presents a significant challenge in practice: conventional models, biased by the overrepresentation of certain high-frequency functional groups, tend to severely overfit to these dominant classes during training Xu et al. (2024). Consequently, they often fail to capture meaningful associations between spectral patterns and less-represented functional groups, leading to worse generalization and the emergence of spurious correlations driven by frequency rather than genuine chemical relevance Haghighatlari et al. (2020); Gallegos et al. (2021). Such limitations hinder model robustness, especially in recognizing rare functional groups that are nonetheless critical for downstream molecular analysis tasks.

Moreover, a molecule should be understood as an integrated whole, composed of multiple intricately interacting substructures Winterbach et al. (2013); Mitra et al. (2013). From a pharmacological standpoint, a molecule can be seen as an organized assembly of co-occurring functional fragments Magura (2008), where functional groups exhibit both statistical and chemical dependencies—i.e., characteristic patterns of co-occurrence Ertl & Schuhmann (2019); Ertl (2017). For instance, hydroxyl and carboxyl groups frequently appear together, as a carboxyl group inherently includes a hydroxyl moiety Cramer et al. (2019); Dimakos & Taylor (2018). Despite this, most existing approaches treat functional group prediction as a multi-label classification problem with independent outputs, thereby completely ignoring these intrinsic chemical relationships. This oversight may result in chemically implausible combinations of functional groups, undermining the validity and reliability of the predicted molecular structures.

In this work, we propose a **Sym**metric Conditional Information Bottleneck (SCIB) framework to seamlessly integrate multi-modal **Spectra** features, named **SymSpectra**, effectively addressing the persistent task of class-imbalanced functional group classification. Built upon information bottleneck principles, the SCIB framework dynamically suppresses redundant cross-modal features while preserving discriminative signals, thereby maximizing spectral complementarity and ensuring prediction stability across diverse spectroscopic conditions. Meanwhile, to address the inherent bias toward underrepresented functional groups caused by data imbalance, SymSpectra innovatively incorporates conditional mutual information (CMI) into the training objective, explicitly prioritizing rare classes. Unlike traditional class reweighting or resampling heuristics, the CMI-guided optimization dynamically quantifies and amplifies the informational significance of spectrally ambiguous and underrepresented functional groups. Furthermore, to holistically model the complex interdependencies among functional groups, we introduce a structured prediction module that processes targets in a predefined order, leveraging earlier predictions as contextual inputs to explicitly encode co-occurrence and exclusivity relationships. This integrated approach substantially improves classification accuracy while ensuring chemically interpretable predictions, particularly for rare or ambiguous functional groups. Evaluated on both simulated and experimental spectroscopic data, SymSpectra achieves a new state-of-the-art F1-score of 0.970, significantly outperforming baseline methods. Notably, the framework demonstrates consistent superiority under challenging imbalance

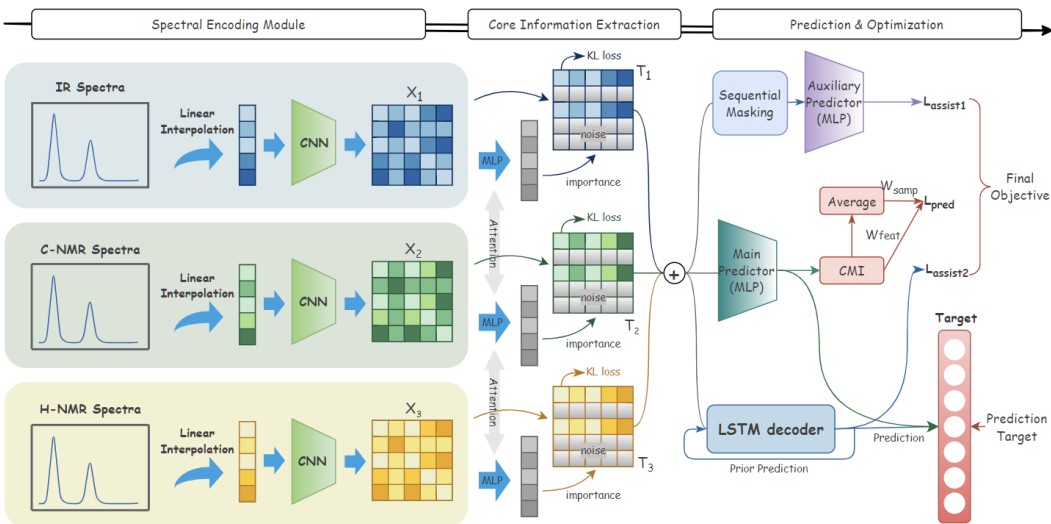

Figure 2: Illustration of the **SymSpectra** model. A SCIB framework integrates multimodal spectroscopic data. During training, three predictors are jointly optimized to compute CMI while capturing functional group dependencies. Only the main predictor and the decoder is used for inference.

scenarios such as label imbalance and structural heterogeneity, exhibiting unmatched robustness and generalizability compared to existing approaches.

## 2 METHODOLOGY

### 2.1 MULTI-MODAL CONDITIONAL INFORMATION BOTTLENECK

Our method focus on extracting task-relevant, non-redundant features from each modality conditioned on the others. This motivates a Symmetric Conditional Information Bottleneck framework, jointly optimizing all modality-specific representations.

**Symmetric Conditional Informational Bottleneck (SCIB)** Given $n$ modalities $\{X^i\}_{i=1}^n$ and target variable $Y$, the optimal representations $\{T^i\}_{i=1}^n$ are obtained by solving:

$$\arg\min_{\{T^i\}} \sum_{i=1}^{n} \Big[ -I(T^i; Y \mid T^{\neg i}) + \beta I(X^i; T^i \mid X^{\neg i}) \Big], \tag{1}$$

where $X^{\neg i}$ and $T^{\neg i}$ denote all other modalities and representations except the i-th one, respectively.

The SCIB framework ensures that each representations $\{T^i\}$:

❶ Preserves unique information about $Y$ not contained in other representations $T^{\neg i}$.

❷ Eliminates redundant information already present in other modalities.

The parameter $\beta \geq 0$ is a Lagrangian multiplier that governs the trade-off between predictive accuracy and representation compression.

### 2.2 MODEL ARCHITECTURE

#### 2.2.1 SPECTRAL ENCODING MODULE

To accommodate the heterogeneous nature of multi-modal spectroscopic data, modality-specific preprocessing strategies are applied, as detailed in Appendix F. Regardless of the modality, each processed spectrum is transformed into a fixed-length vector of 600 dimensions, ensuring dimensional consistency for multimodal integration.

The resulting spectral vector is then passed through two sequential 1D convolutional blocks, each composed of a `Conv1D` layer, followed by `BatchNorm1D`, `ReLU`, and `MaxPool1D`. Formally, for an input vector $X^i \in \mathbb{R}^{600}$, the transformation in each block can be expressed as:

$$O^i = \texttt{MaxPool1D}(\texttt{ReLU}(\texttt{BatchNorm}(\texttt{Conv1D}(X^i)))). \tag{2}$$

This design effectively captures local spectral patterns while robustly promoting consistent feature representations both across different modalities and scales.

### 2.2.2 CORE INFORMATION EXTRACTION

To extract the core representation $T^i$ of each spectrum $X^i$, a unified cross-modal attention mechanism is applied to dynamically estimate the importance of spectral tokens within and across modalities, preserving complementary information while suppressing redundancy. Let the modality-specific representations be $\{O^1, O^2...O^n\} \in \mathbb{R}^{B \times L \times C}$. Firstly, all modalities are concatenated along the sequence dimension to obtain a joint representation:

$$O_{\text{cat}} = \texttt{Linear}(\texttt{Concat}(O^1, O^2...O^n)). \tag{3}$$

Subsequently, one modality (e.g., $O^1$) serves as the query in a multi-head attention mechanism applied over the concatenated sequence, enabling the aggregation of cross-modal context:

$$O^1_{\text{attn}} = \texttt{MultiheadAttention}(Q = O^1, K = O_{\text{cat}}, V = O_{\text{cat}}). \tag{4}$$

The output $O^1_{\text{attn}} \in \mathbb{R}^{B \times L \times C}$ is processed by a feed-forward network in order to compute an importance score for each individual token in the sequence.

$$p_1 = \sigma(\texttt{MLP}(O^1_{\text{attn}})) \in [0, 1]^{B \times L}. \tag{5}$$

The same procedure is applied to $\{O^2...O^n\}$ to obtain $\{p^2...p^n\}$, respectively.

The importance scores learned from cross-modal attention act as soft masks to reweight spectral representations, filtering out task-irrelevant components under the information bottleneck framework. To enforce task-relevant compression, we follow the variational information bottleneck principle Yu et al. (2022) by perturbing the representations with stochastic noise. Specifically, each token representation $H^i_j$ is replaced with Gaussian noise $\epsilon^i \sim \mathcal{N}(\mu_i, \sigma^2_i)$ according to a sampled gate $\lambda^i_j \sim \text{Bernoulli}(p^i_j)$:

$$T^i_j = \lambda^i_j H^i_j + (1 - \lambda^i_j)\epsilon^i. \tag{6}$$

Gumbel-Softmax relaxation is adopt to make the sampling differentiable Maddison et al. (2016):

$$\lambda^i_j = \sigma\left(\frac{1}{t} \log\left(\frac{p^i_j}{1 - p^i_j}\right) + \log\left(\frac{u}{1 - u}\right)\right), \quad u \sim \text{Uniform}\,(0, 1), \tag{7}$$

where $t$ is the temperature parameter set to 1.0. This enables end-to-end optimization of the information bottleneck objective as follows:

$$\min_\theta \beta \sum_{i=1}^n I(X^i; T^i \mid X^{\neg i}). \tag{8}$$

By minimizing the conditional mutual information, the model learns to suppress redundant signals across modalities while retaining discriminative features for classification.

### 2.2.3 DYNAMIC WEIGHTING STRATEGY BASED ON CMI

To exploit spectral complementarity, we extend the information bottleneck with conditional mutual information (CMI). As defined in Equation 1, $I(Y; T^i \mid T^{\neg i})$ quantifies each modality's unique predictive contribution. CMI is approximated via an auxiliary CNN predictor trained under modality dropout: masking $T^i$ and measuring the performance drop in predicting $Y$ from $T^{\neg i}$. This drop proxies $I(Y; T^i \mid T^{\neg i})$, guiding sample- and group-specific weighting in the loss (Section 2.3). Formally, the CMI of $T^i$ and $Y$ given other modality representations $T^{\neg i}$ is:

$$
\begin{aligned}
I(Y; T^i \mid T^{\neg i}) &= \mathbb{E}_{T^i, Y \mid T^{\neg i}}\left[\log \frac{p(Y, T^i \mid T^{\neg i})}{p(Y \mid T^{\neg i})\,p(T^i \mid T^{\neg i})}\right] \\
&= \mathbb{E}_{T^i, Y \mid T^{\neg i}}\left[\log \frac{p(Y \mid T^i, T^{\neg i})\,p(T^i \mid T^{\neg i})}{p(Y \mid T^{\neg i})\,p(T^i \mid T^{\neg i})}\right] \\
&= \mathbb{E}_{T^i, Y \mid T^{\neg i}}\left[\log \frac{p(Y \mid T^i, T^{\neg i})}{p(Y \mid T^{\neg i})}\right],
\end{aligned}
\tag{9}
$$

where $p(Y \mid T^i, T^{\neg i})$ and $p(Y \mid T^{\neg i})$ are estimated separately by the main and auxiliary predictors.

**Loss reweighting with modality-wise and sample-wise weights.** A dual-level dynamic weighting mechanism is introduced to adaptively incorporate these CMI estimates into the training process. Specifically, the total loss function is formally defined as:

$$\mathcal{L}_{\text{total}} = \frac{1}{N} \sum_{i=1}^{N} \left[ W_i^{\text{samp}} \cdot \frac{1}{K} \sum_{j=1}^{K} \ell\left(y_{i,j}, \hat{y}_{i,j}\right) \cdot W_{i,j}^{\text{feat}} \right], \tag{10}$$

In this formulation, $W_i^{\text{samp}}$ is the sample-level weight for instance $i$, and $W_{i,j}^{\text{feat}}$ is the feature-level weight reflecting the relevance of the most informative modality for label $j$. $W_i^{\text{samp}}$ adjusts instance importance, while $W_{i,j}^{\text{feat}}$ refines label-specific predictions based on modality contributions.

**Sample-level weighting.** The sample weight $W_i^{\text{samp}}$ emphasizes samples with higher prediction uncertainty, which typically correspond to challenging functional groups. It is defined as:

$$W_i^{\text{samp}} = \max\left\{ 0, \ 1 + s_1 \cdot \frac{\overline{\text{CMI}}_i - \mu_s}{\sigma_s} \right\}, \tag{11}$$

where $\overline{\text{CMI}}_i$ is the average estimated CMI for all labels in sample $i$, and $\mu_s$, $\sigma_s$ denote the batch mean and standard deviation of $\overline{\text{CMI}}_i$.

**Feature-level weighting.** The feature weight $W_{i,j}^{\text{feat}}$ adjusts the importance of each label based on its modality sensitivity. It is given by:

$$W_{i,j}^{\text{feat}} = \max\left\{ 0, \ 1 + s_2 \cdot \frac{\text{CMI}_{i,j} - \mu_f}{\sigma_f} \right\}, \tag{12}$$

where $\text{CMI}_{i,j}$ denotes the estimated conditional mutual information related to predicting label $j$ for sample $i$, and $\mu_f$, $\sigma_f$ are normalization statistics across all samples and labels.

### 2.2.4 SEQUENTIAL MULTI-LABEL PREDICTION USING LSTM DECODER

Structured label dependencies are modeled using an LSTM decoder that sequentially predicts functional groups from fused multi-modal features. Specifically, given the fused representation $T_f$ obtained from multi-modal spectral inputs, the decoder's initializes its hidden state $h_0$ using a MLP and predicts functional groups sequentially according to a predefined order. At each time step $t$, the previous prediction $y_{t-1}$ is embedded as $e_t$, and the hidden state is updated via:

$$h_t = \text{LSTM}(e_t, h_{t-1}). \tag{13}$$

Predictions are generated via a dropout-regularized linear layer.

To stabilize training and accelerates convergence, Scheduled Sampling Mihaylova & Martins (2019) is used with teacher forcing. At each step, the model uses either the ground truth or its own prediction as input, with the ground truth probability $p$ initialized to 0.5 and decaying exponentially (factor 0.95 per epoch). This gradually shifts reliance to model predictions.

### 2.3 OPTIMIZATION

To jointly learn the model parameters and effectively identify modality-specific core information, we minimize the objective function:

$$\min -I(Y; T^i \mid T^{\neg i}) + \beta I(T^i; X^i \mid X^{\neg i}), \tag{14}$$

where each term corresponds to either prediction or compression. The following sections derive an upper bound for each, which is minimized during training.

### 2.3.1 MINIMIZING $-I(Y; T^i \mid T^{\neg i})$

The first term $-I(Y; T^i \mid T^{\neg i})$ ensures $T^i$ encodes complementary information about $Y$ that is not already captured by $T^{\neg i}$. By the chain rule of mutual information, the objective is decomposed into two sub-components:

$$-I(Y; T^i \mid T^{\neg i}) = -I(Y; T^i, T^{\neg i}) + I(Y; T^{\neg i}), \tag{15}$$

The first term is bounded by the main classification loss. Let $p_\theta(Y \mid T^i, T^{\neg i})$ be a variational predictor; following the variational information bottleneck framework, $-I(Y; T^i, T^{\neg i})$ is approximated using the parametric form:

$$-I(Y; T^i, T^{\neg i}) \leq \mathbb{E}_{Y, T^i, T^{\neg i}} \left[ -\log p_\theta(Y \mid T^i, T^{\neg i}) \right] := \mathcal{L}_{\text{pred}}. \tag{16}$$

The second term $I(Y; T^{\neg i})$ is omitted, as its minimization introduces optimization instability that degrades model performance, as empirically validated in Obs 11. Further details are shown in Appendix E.1.

### 2.3.2   MINIMIZING $I(T^i; X^i \mid X^{\neg i})$

The second term constrains the information transfer from $X^i$ into $T^i$, suppressing modality-specific noise and redundancy. To decompose this term, the chain rule of mutual information is applied:

$$I(T^i; X^i \mid X^{\neg i}) = I(T^i; X^i, X^{\neg i}) - I(T^i; X^{\neg i}). \tag{17}$$

As shown in Appendix E.2, both terms are tractably upper-bounded under Gaussian assumptions using the variational information bottleneck framework. We also explore the impact of different prior distributions in the Appendix G. Specifically:

$$I(T^i; X^i, X^{\neg i}) \leq \mathbb{E}_{X^i, X^{\neg i}} \left[ -\tfrac{1}{2} \log A + \tfrac{1}{2N_i} A + \tfrac{1}{2N_i} B^2 \right] + C, \tag{18}$$

$$-I(T^i; X^{\neg i}) \leq \mathbb{E}_{X^{\neg i}} \left[ -\tfrac{1}{2} \log A' + \tfrac{1}{2N_i} A' + \tfrac{1}{2N_i} (B')^2 \right] + C, \tag{19}$$

where $A$, $B$, $A'$, $B'$ are attention-weighted terms computed from spectral features, $C$ is a constant. Intuitively, minimizing these upper bounds drives the attention mechanism to function as an information filter. The terms $A'$ and $B'$ explicitly penalize the predictability of $T^i$ given the other modalities $X^{\neg i}$. By optimizing this objective, the model learns to assign lower attention weights to features in $X^i$ that are redundant, thereby forcing the representation $T^i$ to focus exclusively on modality-specific, complementary information.

## 3   EXPERIMENT AND ANALYSES

We present experimental results to demonstrate the effectiveness of our model. In this section, we conduct extensive experiments to address the following research questions:

- **RQ1:** Can SymSpectra accurately perform structure elucidation?
- **RQ2:** Can SymSpectra effectively mitigate class imbalance problem?
- **RQ3:** How do individual modules contribute to SymSpectra's performance?

### 3.1   DATASETS AND SETUPS

**Datasets.** We evaluate our model on both simulated and real-world experimental spectroscopic datasets. The simulated dataset from Alberts et al. Alberts et al. (2024) contains 794K molecules with IR, 1H-NMR, $^{13}$C-NMR, and MS/MS spectra. For experimental validation, we curated a dataset of approximately 12K molecules from the Spectral Database for Organic Compounds (SDBS)[1] with corresponding MS, $^{13}$C-NMR, and 1H-NMR spectra, as large-scale multi-modal experimental datasets are not publicly available. Further details are available in Appendix H.

**Baselines.** To thoroughly benchmark our model, we compare it against a diverse set of key architectures, ranging from a specialized 1D-CNN for spectral analysis Jung et al. (2023) and a standard OpenNMT-based Transformer Klein et al. (2018) to the more recent state-of-the-art models proposed by Alberts et al. (2025) and Wu et al. (2025).

**Metrics.** We evaluate model performance using three metrics: sample-level accuracy (ACC), the percentage of samples where all functional groups are correctly predicted; macro-F1, the unweighted average F1-score across all classes, which highlights performance on rare groups; and micro-F1, which calculates the F1-score globally to reflect overall performance. Each experiment is repeated eight times with the same 8:1:1 train/validation/test split. Detailed hyperparameter settings are provided in Appendix B, and a report on time and space consumption is available in Appendix R.

---

[1]https://sdbs.db.aist.go.jp

Table 1: F1-scores for predicting functional groups. The best results, determined by t-tests at a 95% confidence level, are highlighted in **bold**. The second-best results are underlined.

| Spectrum Config. | 1D-CNN | Transformer | Wu et al. | Alberts et al. | SymSpectra |
|---|---|---|---|---|---|
| **Alberts et al. (Simulated Spectra)** | | | | | |
| IR | $\underline{0.895}_{(0.002)}$ | $0.881_{(0.021)}$ | $0.886_{(0.013)}$ | $0.891_{(0.007)}$ | $\mathbf{0.924}_{(0.013)}$ |
| $^{13}$C-NMR | $0.674_{(0.056)}$ | $0.913_{(0.017)}$ | $0.914_{(0.004)}$ | $\underline{0.919}_{(0.012)}$ | $\mathbf{0.921}_{(0.023)}$ |
| $^{1}$H-NMR | $0.839_{(0.005)}$ | $0.935_{(0.031)}$ | $\underline{0.943}_{(0.036)}$ | $\mathbf{0.946}_{(0.027)}$ | $0.925_{(0.007)}$ |
| IR, $^{13}$C-NMR, $^{1}$H-NMR | $0.900_{(0.004)}$ | $0.936_{(0.013)}$ | $0.944_{(0.012)}$ | $\underline{0.947}_{(0.014)}$ | $\mathbf{0.970}_{(0.018)}$ |
| IR, MS/MS$_{pos}$, MS/MS$_{neg}$ | $0.887_{(0.008)}$ | $0.911_{(0.003)}$ | $0.924_{(0.012)}$ | $\underline{0.931}_{(0.031)}$ | $\mathbf{0.949}_{(0.008)}$ |
| **SDBS Database (Experimental Spectra)** | | | | | |
| MS | $0.801_{(0.018)}$ | $0.826_{(0.021)}$ | $\underline{0.837}_{(0.015)}$ | $0.836_{(0.010)}$ | $\mathbf{0.855}_{(0.007)}$ |
| $^{13}$C-NMR | $0.729_{(0.033)}$ | $0.821_{(0.020)}$ | $\underline{0.833}_{(0.014)}$ | $0.836_{(0.011)}$ | $\mathbf{0.849}_{(0.007)}$ |
| $^{1}$H-NMR | $0.701_{(0.027)}$ | $0.779_{(0.025)}$ | $\underline{0.801}_{(0.019)}$ | $\mathbf{0.803}_{(0.018)}$ | $0.796_{(0.008)}$ |
| MS, $^{13}$C-NMR, $^{1}$H-NMR | $0.847_{(0.022)}$ | $0.858_{(0.019)}$ | $0.872_{(0.020)}$ | $\underline{0.881}_{(0.017)}$ | $\mathbf{0.919}_{(0.008)}$ |

## 3.2 MODEL PERFORMANCE (RQ1)

**Obs 1: SymSpectra achieves superior predictive performance over baseline models.** Table 1 presents the predictive performance for molecular structure inference, showing SymSpectra consistently outperforms all baselines across both unimodal and multimodal settings. Furthermore, Figure 3(a) shows SymSpectra significantly improves sample-level accuracy—a key real-world metric. Figure 3(b) shows SymSpectra achieves higher accuracy as the number of functional groups in a molecule increases. This suggests SymSpectra is more robust with challenging samples, likely by extracting and leveraging core substructures to generalize. For practical efficiency, our analysis, detailed in Appendix R, explores the trade-off between modalities and computational cost, leading us to select the three most informative spectra as model input.

**Obs 2: SymSpectra outperforms standard class imbalance handling techniques without requiring external preprocessing.** To rigorously benchmark SymSpectra against established strategies for mitigating class imbalance, we conducted comparative experiments using the simulated dataset. We augmented strong baselines (1D-CNN and Alberts et al.) with three standard techniques: Oversampling (MLSMOTE), Inverse Class Frequency Reweighting, and Focal Loss. As summarized in Table 2, while these heuristic techniques improved baseline performance (e.g., Alberts et al. improved from 0.947 to 0.956 with Focal Loss), SymSpectra still achieved a superior F1-score of 0.970. Statistical analysis (t-test) confirms this advantage is significant across all comparisons ($p < 0.01$), demonstrating that our CMI-based dynamic weighting offers a more effective, data-driven solution to long-tail distributions than static reweighting or resampling heuristics.

**Obs 3: SymSpectra demonstrates robust performance on experimental spectra.** To evaluate real-world performance, we used a dataset of approximately 12,000 molecules with experimental spectra from the SDBS database. As shown in Table 1, while the performance of all models declined on this data, likely due to the limited dataset size and inherent noise and shifts in experimental spectra, SymSpectra exhibited a significantly smaller loss. This resilience is particularly evident in the multimodal setting, where SymSpectra achieves a substantial performance gain with an F1-score of 0.919. This is attributable to its ability to effectively synthesize complementary information from diverse spectral sources. Furthermore, in Appendix I we analyzed various data augmentation techniques and achieved additional performance improvements. Crucially, the ability to recover performance through realistic perturbations like horizontal shifts and noise suggests that the observed Sim-to-Real gap is primarily a result of domain shifts that can be mitigated, rather than a fundamental defect in the model architecture. These results, along with simulations of spectral noise and missing modalities detailed in Appendix J, further confirm the model's robustness.

**Obs 4: SymSpectra retains more information for samples with rich functional groups, balancing retention and compression.** To examine how SymSpectra adapts its information allocation based on molecular complexity, we analyzed the relationship between functional group count and

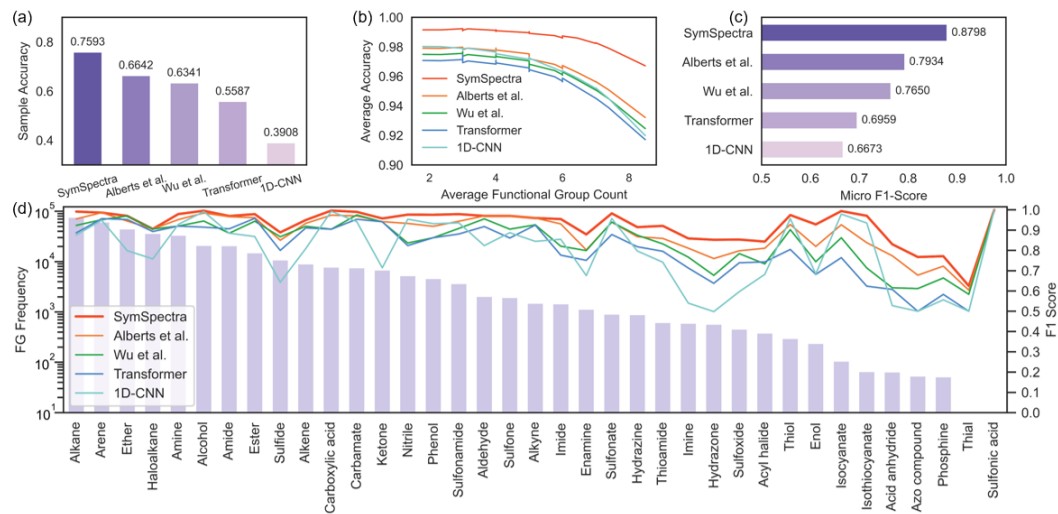

Figure 3: (a) Model-wise sample-level accuracy; (b) Accuracy across molecules with varying functional group counts; (c) Model Performance on Structurally Disjoint Test Set; (d) Correlation between functional group count and macro-F1 score.

modality-wise importance distributions[2]. The distribution of importance scores in Appendix M demonstrate that molecules with richer functional groups consistently receive higher importance scores, indicating SymSpectra preserves more information for complex inputs. This adaptive mechanism effectively tailors the information processing strategy to each sample's inherent complexity.

### 3.3 CAPABILITY TO MITIGATE DATA IMBALANCE (RQ2)

**Obs 5: SymSpectra effectively mitigates class imbalance by achieving superior performance on low-frequency functional groups.** To evaluate SymSpectra's robustness under class imbalance, we assessed its performance on functional groups with varying frequencies in the test set. As shown in Figure 3 (d), our model consistently outperforms the baseline across all categories. The advantage is particularly notable for low-frequency classes such as *Acid anhydride* and *Azo compound*, where the baseline yields near-zero recall. This indicates that our model better captures informative patterns from limited data, enhancing generalization.

**Obs 6: SymSpectra effectively mitigates sample imbalance while ensuring stable performance across varying training conditions.** To evaluate SymSpectra's performance under extreme class imbalance, we selected four most representative rare functional groups as minority classes and progressively reduced their positive training samples until reaching 25% of the original count. All models were trained on these datasets and evaluated on a fixed test set. As shown in Figure 4, although all models demonstrate improved F1 scores for minority classes as the imbalance level decreases, the performance advantage of SymSpectra becomes increasingly pronounced as the imbalance severity reduces, highlighting its greater capacity to fully utilize minority samples.

**Obs 7: SymSpectra effectively mitigates structural imbalance, enabling robust generalization to unseen structures.** To rigorously evaluate model generalization, we employed the scaffold-based clustering strategy detailed in Appendix K. This method partitions the training and test sets into structurally distinct regions of chemical space and thus creates a significant structural imbalance. As shown in Figure 3(c), this intentionally challenging scenario led to a noticeable performance drop across virtually all baselines. Despite this difficulty, SymSpectra achieved a remarkable F1-score of

---

[2]To generate the visualization in Figure 8, 500 molecules were randomly selected and ranked by functional group count; the top 5 (hardest) and bottom 5 (easiest) samples

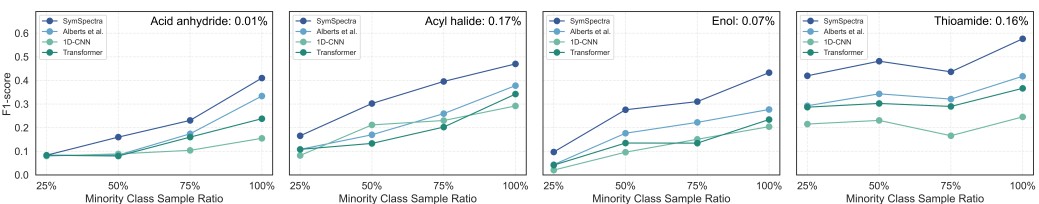

Figure 4: Prediction F1 scores for four functional groups under varied training-set configurations. Each model is independently trained per setting and evaluated on a shared test set. Functional groups and their positive ratios appear in each subfigure's top-right corner.

Table 2: Comparison of SymSpectra against baselines enhanced with standard class imbalance mitigation strategies. P-values indicate the statistical significance of the performance gap compared to SymSpectra.

| Model | Training Strategy | F1-Score | $p$-value (vs. SymSpectra) |
|---|---|---|---|
| **SymSpectra** | **Original** | **0.970** | - |
| Alberts et al. | + Focal Loss | 0.956 | $8.13 \times 10^{-3}$ |
| | + Oversampling | 0.954 | $1.89 \times 10^{-3}$ |
| | + Class Reweighting | 0.951 | $3.17 \times 10^{-3}$ |
| | Original | 0.947 | $9.29 \times 10^{-4}$ |
| 1D-CNN | + Focal Loss | 0.917 | $6.82 \times 10^{-5}$ |
| | + Class Reweighting | 0.915 | $2.41 \times 10^{-4}$ |
| | + Oversampling | 0.910 | $3.09 \times 10^{-5}$ |
| | Original | 0.900 | $8.15 \times 10^{-6}$ |

0.8798, significantly outperforming all other competing models tested. This result strongly indicates SymSpectra's superior ability to handle structural imbalance, ensuring stable and high performance even when predicting diverse and structurally novel molecules.

### 3.4 Ablation Study and Sensitivity Analysis (RQ3)

**Obs 8: The prediction order of functional groups significantly influences model performance.** To investigate the impact of prediction order on model performance, we evaluate several strategies against a non-sequential CNN, including predefined orders based on IUPAC nomenclature priorities Jenkins et al. (1991) and data-driven orders learned via mutual information or GNN. As shown in Table 4, sequential prediction significantly improves performance on functional groups where the baseline struggles, highlighting the importance of modeling inter-label dependencies. The GNN-derived and IUPAC-based orders were most effective, likely by capturing higher-order chemical dependencies. The impact of order is significant; for instance, predicting *Imines* early yields poor results, while *Aldehydes* benefit from later placement that leverages contextual cues. Consequently, we adopt IUPAC order to ensure generalizability, as this predefined convention is immune to dataset bias. Full details on ordering strategies and decoder analysis are available in Appendix L and P.

**Obs 9: The dynamic weighting strategy is crucial for mitigating data imbalances.** To assess the contribution of each component, we performed an ablation study. As shown in Table 3, removing the dynamic weighting mechanism caused a performance drop across all tiers, with the largest decline in low-frequency groups. Visualization in Appendix N shows that samples with more functional groups or harder-to-predict categories receive higher

Table 3: Ablation study of model components, showing average F1-scores across functional group frequencies.

| Component | High | Medium | Low |
|---|---|---|---|
| SymSpectra | 0.967 | 0.942 | 0.866 |
| - Dynamic Weight | 0.957 | 0.922 | 0.844 |
| - SCIB Compression | 0.949 | 0.919 | 0.827 |
| - LSTM Decoder | 0.966 | 0.933 | 0.854 |

Table 4: Prediction results for five representative functional groups under different label ordering strategies. Indices correspond to functional groups listed in the footnote.[4] Best results are in **bold**, second-best are underlined.

| Category | Order | Relative Sequence | 8 | 9 | 13 | 16 | 22 |
|---|---|---|---|---|---|---|---|
| Predefined | IUPAC | [13, 16, 8, 9, 22] | **0.942** | 0.926 | 0.974 | 0.904 | **0.885** |
| Data-driven | Mutual Info | [16, 13, 9, 8, 22] | 0.930 | 0.826 | 0.877 | 0.896 | 0.872 |
| | GNN | [22, 13, 16, 8, 9] | 0.883 | **0.929** | 0.985 | **0.909** | 0.756 |
| Non-sequential | CNN | no order | 0.884 | 0.927 | **0.992** | 0.868 | 0.540 |

weights. By adaptively increasing the weights of underrepresented and difficult groups based on CMI, the model prevents these classes from being overlooked. In addition, removing SCIB compression or the LSTM decoder degraded performance, especially on low-frequency groups, highlighting the complementary role of all components.

**Obs 10: $\beta$ measures information compression, while $s_1$ and $s_2$ control the focus on challenging samples and functional groups.** We performed a sensitivity analysis of $\beta$, $s_1$, and $s_2$ to assess their impact on model performance. According to Equation 1, $\beta$ balances information compression and prediction accuracy, while $s_1$ and $s_2$ (Equations 11 and 12) regulate the emphasis on complex samples. As shown in Appendix Q, setting $s_1 = s_2 = 0.3$ yields optimal performance, as larger values overly emphasize difficult samples and smaller values fail to highlight critical ones. Similarly, $\beta = 1e-6$ gives the best result, since higher values discard meaningful representations via excessive compression, while lower values miss essential substructures.

**Obs 11: Minimizing $I(Y; T^{\neg i})$ negatively impacts final prediction performance.**
The term $I(Y; T^{\neg i})$ quantifies the information about the target $Y$ contained in the context modalities $T^{\neg i}$. Minimizing this term essentially forces the context representations to be uninformative about the label, which creates a fundamental conflict with the main predictor's

Table 5: Performance impact of including the conflicting term $I(Y; T^{\neg i})$.

| Weight ($\lambda$) | 0 (Ours) | 0.01 | 0.1 | 1.0 |
|---|---|---|---|---|
| F1-Score | **0.970** | 0.956 | 0.952 | 0.931 |

goal of maximizing the joint mutual information for accurate classification. To empirically verify this, we conducted an experiment by incorporating the upper bound of this term into the training objective, controlled by a weight hyperparameter $\lambda$. As shown in Table 5, increasing the weight of this penalty term leads to a monotonic deterioration in model performance. This finding aligns with the Variational Information Bottleneck (VIB) framework , which observes similar optimization instability with conflicting objectives. Following their implementation strategy, we exclude this term from our final objective to ensure optimal predictive performance.

## 4 CONCLUSION

In this work, we present **SymSpectra**, a novel framework that integrates multimodal spectral data via a Symmetric Conditional Information Bottleneck framework. Specifically, to address label imbalance, conditional mutual information is incorporated into the training objective, while a dedicated module captures dependencies among functional groups. Experiments on benchmark spectral datasets demonstrate state-of-the-art performance in molecular structure recognition, with significant improvements in identifying rare functional groups. Our approach exhibits strong robustness across various imbalanced scenarios and generates predictions that align more closely with chemical reasoning, thereby supporting downstream tasks such as drug discovery and material design.

---

[4]8: Amide, 9: Amine, 13: Carboxylic acid, 16: Ester, 22: Imine.

## 5 REPRODUCIBILITY

We provide the complete implementation in the repository along with guidance on how to reproduce our results. Our code is available at `https://anonymous.4open.science/r/SymSpectra-0017`.

## 6 ETHICS STATEMENT

Our study does not involve human participants, personal data, or sensitive information. The datasets and resources used are either publicly available or released under appropriate licenses. We confirm that our research does not raise any ethical concerns related to privacy, safety, fairness, or potential misuse. The contributions of this work are intended solely for advancing scientific research and are not designed or evaluated for harmful applications.

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

## A    USE OF LARGE LANGUAGE MODELS (LLMs)

A large language model (LLM) was employed exclusively for writing assistance and text refinement, including grammar correction, stylistic adjustments, and improving conciseness. The LLM did not contribute to research design, data analysis, model development, or interpretation of results. All technical content, experiments, and conclusions were entirely conceived, conducted, and validated by the authors.

## B    TRAINING SETTINGS

Our model was trained for 100 epochs on a single NVIDIA A100 GPU (80GB) with a batch size of 128. We employed the Adam optimizer with a learning rate of 3e-4, managed by a cosine annealing scheduler. An LSTM-based decoder with Scheduled Sampling was utilized for the sequential prediction task. Based on hyperparameter tuning, the information bottleneck coefficient $\beta$ was set to 1e-6, and the sample emphasis regulators, $s_1$ and $s_2$, were both set to 0.3. The total training time was approximately 7 hours.

## C    RELATED WORK

### C.1    SPECTROSCOPY-BASED MOLECULAR MODELING

Recent advances in machine learning have significantly advanced molecular modeling from spectroscopic data. Devata et al. Devata et al. (2024) introduced DeepSPInN, a reinforcement learning approach that predicts molecular structures directly from infrared and $^{13}$C-NMR spectra, without relying on spectral databases. Kim et al. Kim et al. (2023) developed DeepSAT, a neural network-based system that extracts structural features from $^1$H-$^{13}$C HSQC NMR spectra to assist molecular structure annotation. Baygi et al. Baygi & Barupal (2024) proposed IDSL_MINT, a transformer-based framework that translates MS/MS spectra into molecular fingerprint descriptors to enhance annotation in untargeted metabolomics and exposomics. In parallel, Stravs et al. Stravs et al. (2022) explored the use of recurrent neural networks for de novo molecular structure generation from mass spectrometry data. More recently, diffusion models and Large Language Models have also been adapted for spectral analysis. DiffMS Bohde et al. (2025) utilizes diffusion models to generate molecular graphs conditioned on mass spectra, employing combinatorial optimization to embed spectral peaks. Similarly, DiffSpectra Wang et al. (2025) investigates the joint modeling of 2D and 3D molecular structures, enabling the prediction of conformations from multi-modal spectra. In the domain of general-purpose models, a comprehensive multimodal benchmark Guo et al. (2024) has been established to evaluate the capabilities of LLMs in solving molecular puzzles, specifically focusing on their potential in spectrum interpretation and molecule construction. While these methods demonstrate considerable progress, most focus exclusively on mass spectrometry and often overlook the integration of diverse spectroscopic modalities. Furthermore, mass spectrometry remains expensive, noise-sensitive, and difficult to standardize in automated workflows. To address these challenges, Alberts et al. Alberts et al. (2024) introduced a multimodal spectroscopic dataset comprising 790,000 entries, enabling joint analysis across multiple spectroscopic techniques. Their reference models for structural inference and functional group classification establish a strong baseline for future research. Together, these contributions highlight both the potential and limitations of current approaches, motivating our development of an integrative, multi-spectroscopic framework for robust and generalizable molecular structure analysis.

### C.2    INFORMATION BOTTLENECK THEORY

The Information Bottleneck (IB) framework offers a systematic approach to distilling concise yet meaningful representations from intricate datasets, which is particularly valuable for tasks such as noise reduction and data compression. Building upon this, PGIB Yu et al. (2020) generalizes the IB principle by incorporating a mutual information estimation mechanism tailored for non-uniform graph structures, as well as introducing a connectivity-based loss to enhance the robustness of information extraction. VGIB Yu et al. (2022) advances this direction by injecting Gaussian perturbations into node embeddings, thereby moderating the transfer of information between the original and mod-

ified graphs and further enhancing stability. Additionally, Lee et al. Lee et al. (2023) expanded the graph information bottleneck to the field of molecular relational learning, proposing the Conditional Graph Information Bottleneck (CGIB) theory, which aims to retain as much relevant information as possible with paired graphs while obtaining compressed subgraphs. The CGIB theory addresses the issue of extracting independent subgraphs in GIB for MRL tasks, but considering all information from another graph during interaction can introduce excessive noise.

# D  BROADER IMPACTS AND LIMITATION

## D.1  BROADER IMPACTS

This work enables robust prediction of functional group presence under imbalanced scenarios, aligning closely with chemical reasoning. By detecting rare functional groups, it supports early drug screening, as such substructures often signal unique bioactivity. Additionally, **SymSpectra** models functional group interactions to enhance interpretability and consistency, offering valuable insights for materials science and diagnostics.

## D.2  LIMITATION

Despite its strengths, SymSpectra still has limitations. First, its generalization is constrained by the diversity of the training data, leading to suboptimal performance on rare or novel functional groups. Second, resolving severe spectral overlap in complex molecules remains a significant hurdle, which can cause ambiguity in feature attribution. The framework also identifies the presence of functional groups but not their connectivity, making it difficult to distinguish between certain structural isomers. Addressing these challenges will require more diverse datasets and architectural enhancements for more fine-grained structural elucidation.

# E  PROOF

## E.1  MINIMIZING $-I(Y; T^i \mid T^{\neg i})$

The objective of minimizing the conditional mutual information $-I(Y; T^i \mid T^{\neg i})$ is to encourage the representation $T^i$ to capture as much discriminative information as possible about the label $Y$, conditioned on the remaining representations $T^{\neg i}$. By applying the chain rule of mutual information, we decompose this term as:

$$-I(Y; T^i \mid T^{\neg i}) = -I(Y; T^i, T^{\neg i}) + I(Y; T^{\neg i}), \tag{20}$$

where $I(Y; T^i, T^{\neg i})$ denotes the mutual information between $Y$ and the joint representation $(T^i, T^{\neg i})$, while $I(Y; T^{\neg i})$ captures the information about $Y$ in the remaining representations.

### E.1.1  MINIMIZING $-I(Y; T^i, T^{\neg i})$

To minimize $-I(Y; T^i, T^{\neg i})$, we derive an upper bound using the negative log-likelihood, resulting in the supervised prediction loss:

$$
\begin{aligned}
-I(Y; T^i, T^{\neg i}) &= \mathbb{E}_{(T^i, T^{\neg i}, Y)} \left[ \log \frac{p(Y)}{p(Y \mid T^i, T^{\neg i})} \right] \\
&\leq \mathbb{E}_{(T^i, T^{\neg i}, Y)} \left[ -\log p_\theta(Y \mid T^i, T^{\neg i}) \right] \\
&= -\mathbb{E}_{(T^i, T^{\neg i}, Y)} \left[ \log p_\theta(Y \mid T^i, T^{\neg i}) \right] \\
&= \mathcal{L}_{\text{pred}},
\end{aligned}
\tag{21}
$$

where $p_\theta(Y \mid T^i, T^{\neg i})$ is a probabilistic predictor parameterized by $\theta$. It infers the target label $Y$ based on the full set of spectral representations $(T^i, T^{\neg i})$. Therefore, minimizing the primary prediction loss $\mathcal{L}_{\text{pred}}(Y, T^i, T^{\neg i})$, often implemented as a cross-entropy loss, serves as a tractable surrogate for minimizing $-I(Y; T^i, T^{\neg i})$.

### E.1.2 Minimizing $I(Y; T^{\neg i})$

The second term in the decomposition, $I(Y; T^{\neg i})$, represents the mutual information that the set of all other modalities, $T^{\neg i}$, shares with the target $Y$. However, minimizing the mutual information $I(Y; T^{\neg i})$ requires forcing the representations $T^{\neg i}$ to become uninformative about $Y$. This objective is fundamentally contradictory to the primary model's goal of learning meaningful features by minimizing its own prediction loss. Such a setup introduces an unstable, adversarial training dynamic that can impede convergence and degrade the quality of the learned representations. Given this theoretical conflict and the practical instability it creates, we omit this term from our final objective.

### E.2 Minimizing $I(T^i; X^i \mid X^{\neg i})$

The goal of minimizing $I(T^i; X^i \mid X^{\neg i})$ is to compress the information from $X^i$ as much as possible, conditioned on $X^{\neg i}$. Using the chain rule of mutual information, we express this as:

$$I(T^i; X^i \mid X^{\neg i}) = I(T^i; X^i, X^{\neg i}) - I(T^i; X^{\neg i}). \tag{22}$$

Minimizing $I(T^i; X^i, X^{\neg i})$ encourages the encoder to extract only the most essential information from the combined inputs, while minimizing $-I(T^i; X^{\neg i})$ ensures that $T^i$ retains as much dependence on $X^{\neg i}$ as needed to support meaningful disentanglement.

### E.2.1 Minimizing $I(T^i; X^i, X^{\neg i})$

To upper-bound the mutual information $I(T^i; X^i, X^{\neg i})$, we adopt a variational approximation approach inspired by the Variational Autoencoder (VAE) framework. Specifically, we introduce a variational distribution $q(T^i)$ to approximate the marginal $p(T^i)$:

$$
\begin{aligned}
I(T^i; X^i, X^{\neg i}) &= \mathbb{E}_{p(X^i, X^{\neg i}, T^i)} \left[ \log \frac{p(T^i \mid X^i, X^{\neg i})}{p(T^i)} \right] \\
&= \mathbb{E}_{p(X^i, X^{\neg i}, T^i)} \left[ \log \frac{p(T^i \mid X^i, X^{\neg i})}{q(T^i)} \cdot \frac{q(T^i)}{p(T^i)} \right] \\
&= \mathbb{E}_{p(X^i, X^{\neg i}, T^i)} \left[ \log \frac{p(T^i \mid X^i, X^{\neg i})}{q(T^i)} \right] - \mathbb{E}_{p(X^i, X^{\neg i}, T^i)} \left[ \log \frac{q(T^i)}{p(T^i)} \right] \\
&= \mathbb{E}_{p(X^i, X^{\neg i}, T^i)} \left[ \log \frac{p(T^i \mid X^i, X^{\neg i})}{q(T^i)} \right] - D_{\mathrm{KL}}(p(T^i) \| q(T^i)) \\
&\leq \mathbb{E}_{p(X^i, X^{\neg i}, T^i)} \left[ \log \frac{p(T^i \mid X^i, X^{\neg i})}{q(T^i)} \right].
\end{aligned}
\tag{23}
$$

The KL divergence is non-negative, providing a valid upper bound.

Following the Variational Information Bottleneck (VIB) principle, we model $q(T^i)$ as a Gaussian perturbed version of the encoder's output, where the noise is sampled as $\epsilon \sim \mathcal{N}(\mu_{T^i}, \sigma_{T^i}^2)$. Let the representation $T^i$ be aggregated (e.g., via summation) over local spectral features. By the additive property of Gaussian distributions, the aggregated representation is also Gaussian:

$$q(T^i) = \mathcal{N}(N_i \mu_{T^i}, N_i \sigma_{T^i}^2). \tag{24}$$

For the conditional distribution $p(T^i \mid X^i, X^{\neg i})$, we assume a Gaussian with mean shifted by attention-weighted local feature deviations:

$$p(T^i \mid X^i, X^{\neg i}) = \mathcal{N}\left( N_i \mu_{T^i} + \sum_{j=1}^{N_i} \lambda_j (h_j - \mu_{T^i}), \sum_{j=1}^{N_i} (1 - \lambda_j)^2 \sigma_{T^i}^2 \right). \tag{25}$$

Substituting into the mutual information bound yields:

$$I(T^i; X^i, X^{\neg i}) \leq \mathbb{E}_{X^i, X^{\neg i}} \left[ -\frac{1}{2} \log A + \frac{1}{2N_i} A + \frac{1}{2N_i} B^2 \right] + C, \tag{26}$$

where $A = \sum_{j=1}^{N_i} (1 - \lambda_j)^2$, $B = \sum_{j=1}^{N_i} \lambda_j \frac{(h_j - \mu_{T^i})}{\sigma_{T^i}}$, and $C$ is a constant.

### E.2.2 MINIMIZING $-I(T^i; X^{\neg i})$

To upper-bound $-I(T^i; X^{\neg i})$, we again use a variational approximation:

$$
\begin{aligned}
-I(T^i; X^{\neg i}) &= -\mathbb{E}_{p(T^i, X^{\neg i})}\left[\log \frac{p(T^i \mid X^{\neg i})}{p(T^i)}\right] \\
&= -\mathbb{E}_{p(T^i, X^{\neg i})}\left[\log \frac{p(T^i \mid X^{\neg i})}{q(T^i)}\right] + D_{\mathrm{KL}}(q(T^i)\|p(T^i)) \\
&\leq -\mathbb{E}_{p(T^i, X^{\neg i})}\left[\log \frac{p(T^i \mid X^{\neg i})}{q(T^i)}\right],
\end{aligned}
\tag{27}
$$

where the KL term is non-negative and thus may be omitted during optimization.

We retain the Gaussian assumptions:

$$
q(T^i) = \mathcal{N}(N_i \mu_{T^i}, N_i \sigma_{T^i}^2),
\tag{28}
$$

$$
p(T^i \mid X^{\neg i}) = \mathcal{N}\left(N_i \mu_{T^i} + \sum_{j=1}^{N_i} \lambda_j^{(\neg i)}(h_j^{(\neg i)} - \mu_{T^i}), \sum_{j=1}^{N_i}(1 - \lambda_j^{(\neg i)})^2 \sigma_{T^i}^2\right),
\tag{29}
$$

where $\lambda_j^{(\neg i)}$ and $h_j^{(\neg i)}$ are attention weights and local features derived from $X^{\neg i}$ only.

Thus, we have:

$$
-I(T^i; X^{\neg i}) \leq \mathbb{E}_{X^{\neg i}}\left[-\frac{1}{2}\log A' + \frac{1}{2N_i}A' + \frac{1}{2N_i}(B')^2\right] + C,
\tag{30}
$$

where

$$
A' = \sum_{j=1}^{N_i}(1 - \lambda_j^{(\neg i)})^2, \quad B' = \sum_{j=1}^{N_i} \lambda_j^{(\neg i)}\frac{(h_j^{(\neg i)} - \mu_{T^i})}{\sigma_{T^i}},
\tag{31}
$$

and $C$ is a constant that does not affect optimization.

This regularization term encourages $T^i$ to remain conditionally independent of the unrelated modality $X^{\neg i}$, thereby promoting disentangled and modality-specific representation learning.

## F PREPROCESSING OF MULTIMODAL SPECTRAL DATA

To better exploit the unique characteristics of each spectral modality, we explored and empirically selected distinct preprocessing strategies tailored to the data characteristics of each type. The preprocessing procedures for each modality are described as follows.

### F.1 PREPROCESSING OF $^1$H-NMR AND $^{13}$C-NMR SPECTRA

All NMR spectra are first linearly interpolated to a fixed length of 600 to ensure consistent input dimensions. Given two adjacent sampling points $(x_i, y_i)$ and $(x_{i+1}, y_{i+1})$, the interpolated value at position $x \in [x_i, x_{i+1}]$ is computed as:

$$
y(x) = y_i + \frac{(x - x_i)}{(x_{i+1} - x_i)}(y_{i+1} - y_i).
\tag{32}
$$

After interpolation, min-max normalization is applied to scale each spectrum into the range $[0, 1]$:

$$
x_i^{\mathrm{norm}} = \frac{x_i - \min(x)}{\max(x) - \min(x)},
\tag{33}
$$

where $x_i$ denotes the $i$-th interpolated intensity value, and $\min(x)$, $\max(x)$ are the minimum and maximum values of the spectrum, respectively.

## F.2 PREPROCESSING OF IR SPECTRA

Infrared (IR) spectra undergo the same interpolation process to 600 points as the NMR data (see Eq. 32) to maintain consistent input dimensionality. However, we omit normalization for IR spectra, as empirical results suggest that removing absolute intensity information can degrade performance. Unlike NMR, absolute peak heights in IR spectra often encode discriminative cues related to functional group presence or bond strength. Thus, the interpolated raw signals are used directly as model input without additional scaling.

## F.3 PREPROCESSING OF $MS/MS_{pos}$ AND $MS/MS_{neg}$ SPECTRA

Raw MS/MS spectra are represented as a set of discrete fragment peaks, each described by a tuple $(m/z_i, I_i)$, where $m/z_i \in \mathbb{R}^+$ is the mass-to-charge ratio of the $i$-th fragment ion and $I_i \in \mathbb{R}^+$ is its corresponding intensity. These spectra are inherently sparse and vary in length across samples. To enable uniform input for batch processing and model training, we convert each spectrum into a fixed-length continuous representation through a two-step process: Gaussian-based smoothing of fragment peaks followed by cubic spline interpolation over a predefined $m/z$ grid.

### F.3.1 STEP1: GAUSSIAN DIFFUSION ENCODING

Firstly, each spectrum is encoded into a fixed-length vector of 6000 bins, corresponding to the $m/z$ range $[0, 600)$ with a resolution of 0.1. For each peak $(m/z_i, I_i)$, its corresponding position in the vector space is computed as

$$p_i = \lfloor 10 \cdot m/z_i \rfloor, \tag{34}$$

where $p_i \in \{0, 1, \ldots, 5999\}$. To account for small shifts in $m/z$ values and experimental noise, a Gaussian diffusion kernel centered at $p_i$ with standard deviation $\sigma$ is applied as:

$$G_i(x) = \frac{1}{Z_i} \cdot \exp\left(-\frac{(x - p_i)^2}{2\sigma^2}\right), \tag{35}$$

where $Z_i = \sum_x \exp\left(-\frac{(x-p_i)^2}{2\sigma^2}\right)$ is a normalization factor ensuring $\sum_x G_i(x) = 1$. The spectral vector $\mathbf{s} \in \mathbb{R}^{6000}$ is then constructed by accumulating the weighted contributions of all peaks:

$$\mathbf{s}[x] = \sum_{i=1}^{N} I_i \cdot G_i(x), \quad x \in \{0, \ldots, 5999\}. \tag{36}$$

This step produces a smooth, dense representation that retains the shape and intensity of each peak while mitigating resolution sensitivity.

### F.3.2 STEP2: SPLINE-BASED DOWNSAMPLING

To reduce computational cost while preserving the overall shape of the spectrum, we compress the 6000-dimensional vector $\mathbf{s} \in \mathbb{R}^{6000}$ into a 600-dimensional vector $\tilde{\mathbf{s}} \in \mathbb{R}^{600}$ using cubic spline interpolation. Specifically, a smooth curve $f(x)$ is fitted to $\mathbf{s}$, and $\mathbf{s}$ is then sampled at 600 evenly spaced positions over the same range:

$$\tilde{\mathbf{s}}[j] = \max\left(0, f(x_j)\right), \quad j = 1, \ldots, 600. \tag{37}$$

Here, negative values from interpolation are clipped to zero to maintain non-negativity. Finally, $\ell_2$ normalization is applied:

$$\tilde{\mathbf{s}} \leftarrow \frac{\tilde{\mathbf{s}}}{\|\tilde{\mathbf{s}}\|_2 + \varepsilon}, \tag{38}$$

where $\varepsilon$ is a small constant to avoid division by zero. This step produces a compact and scale-invariant spectral representation suitable for model input.

This preprocessing pipeline addresses three key challenges of raw MS/MS spectra: sparsity, variable length, and sensitivity to $m/z$ shifts. The Gaussian diffusion step produces a stable, smooth spectrum, while the spline-based downsampling reduces dimensionality with minimal information loss. The resulting fixed-length, normalized vector $\tilde{\mathbf{s}} \in \mathbb{R}^{600}$ serves as a robust input to downstream models.

## G    IMPACT OF LATENT PRIOR DISTRIBUTION

### G.1    COMPARATIVE ABLATION STUDY

The selection of the prior distribution for the latent variables, $q(t_m)$ and $q(t_a)$, is a key hyperparameter in our proposed framework. To determine the optimal choice, we conduct an ablation study comparing three canonical distributions: the standard normal $\mathcal{N}(0, I)$, the Laplace$(0, 1)$, and the Gamma$(k = 1, \theta = 1)$. The Laplace prior is selected for its tendency to induce sparsity, while the Gamma prior enforces non-negativity in the latent space.

The results of this analysis are summarized in Table 6. The model configured with a Gaussian prior demonstrates superior performance, achieving the highest F1-score. This suggests that the unimodal and symmetric properties of the Gaussian distribution provide a well-suited inductive bias for compressing multi-modal spectral information into a flexible and effective latent representation. In contrast, the sparsity induced by the Laplace prior or the non-negativity constraint of the Gamma prior appear to be overly restrictive for this task. Consequently, we adopt the $\mathcal{N}(0, I)$ prior for all other experiments presented in this work.

Table 6: Impact of different latent prior distributions on functional group classification performance. All other hyperparameters are held constant during this ablation.

| Prior Distribution | F1-Score |
|---|---|
| Gaussian $\mathcal{N}(0, I)$ | **0.970** |
| Laplace$(0, 1)$ | 0.958 |
| Gamma$(k = 1, \theta = 1)$ | 0.951 |

### G.2    VALIDATION OF THE GAUSSIAN ASSUMPTION

Complementing the comparative experiments above, we further clarify that the choice of a Gaussian distribution for our representations is theoretically grounded in the Central Limit Theorem. Specifically, our representation is formed by aggregating a large number of local features, a process that includes pooling and attention mechanisms (Equations 2, 3, and 4). According to the Central Limit Theorem, when aggregating a large number of independent or weakly dependent random variables, the resulting distribution naturally converges to a Gaussian distribution, regardless of the original distribution of the individual features.

To empirically validate this assumption, we conducted tests on the feature distributions of the three spectral modalities (IR, $^{13}$C-NMR, and $^{1}$H-NMR). We tested the compressed feature distributions for these spectra using the Shapiro-Wilk and Kolmogorov-Smirnov (K-S) tests for normality.

Table 7: Normality test results (p-values) for compressed feature distributions across spectral modalities. High p-values ($> 0.05$) indicate that the hypothesis of normality cannot be rejected.

| Test Method | IR | $^{13}$C-NMR | $^{1}$H-NMR |
|---|---|---|---|
| Shapiro-Wilk | 0.1648 | 0.1199 | 0.1479 |
| Kolmogorov-Smirnov | 0.5610 | 0.4210 | 0.4715 |

As shown in Table 7, the p-values from both tests are consistently greater than the significance level of 0.05. These results suggest that the distributions of the compressed features for all three spectral types align with the Gaussian distribution assumption, thereby justifying our choice of prior.

## H    FUNCTIONAL GROUPS DEFINITION

Functional groups serve as the foundational building blocks that impart specific chemical behaviors and biological activities to organic molecules. In this work, we leverage a curated library of chemi-

cally significant substructures, computationally encoded using SMARTS patterns, to achieve precise and interpretable molecular characterization at the subgraph level. These structural descriptors capture key functional motifs and enable efficient substructure recognition through graph isomorphism-based methods, ensuring scalability to large datasets.

The functional groups considered in our analysis are listed in Table 8, covering a diverse array of chemically and pharmacologically relevant units—such as hydroxyl (-OH), carbonyl (C=O), and amino ($-NH_2$) groups. Functional group identification is conducted using cheminformatics libraries such as RDKit, which support efficient substructure searching across molecular graphs. This strategy facilitates the extraction of chemically meaningful features and supports downstream tasks including molecular property prediction, reactivity analysis, and structure-based clustering.

Table 8: Predefined Functional Groups and Their SMARTS Patterns

| No. | Functional Group | SMARTS Pattern |
|---|---|---|
| 1 | Acid anhydride | `[CX3](=[OX1])[OX2][CX3](=[OX1])` |
| 2 | Acyl halide | `[CX3](=[OX1])[F,Cl,Br,I]` |
| 3 | Alcohol | `[#6][OX2H]` |
| 4 | Aldehyde | `[CX3H1](=O)[#6,H]` |
| 5 | Alkane | `[CX4;H3,H2]` |
| 6 | Alkene | `[CX3]=[CX3]` |
| 7 | Alkyne | `[CX2]#[CX2]` |
| 8 | Amide | `[NX3][CX3](=[OX1])[#6]` |
| 9 | Amine | `[NX3;H2,H1,H0;!$(NC=O)]` |
| 10 | Arene | `[cX3]1[cX3][cX3][cX3][cX3][cX3]1` |
| 11 | Azo compound | `[#6][NX2]=[NX2][#6]` |
| 12 | Carbamate | `[NX3][CX3](=[OX1])[OX2H0]` |
| 13 | Carboxylic acid | `[CX3](=O)[OX2H]` |
| 14 | Enamine | `[NX3][CX3]=[CX3]` |
| 15 | Enol | `[OX2H][#6X3]=[#6]` |
| 16 | Ester | `[#6][CX3](=O)[OX2H0][#6]` |
| 17 | Ether | `[OD2]([#6])[#6]` |
| 18 | Haloalkane | `[#6][F,Cl,Br,I]` |
| 19 | Hydrazine | `[NX3][NX3]` |
| 20 | Hydrazone | `[NX3][NX2]=[#6]` |
| 21 | Imide | `[CX3](=[OX1])[NX3][CX3](=[OX1])` |
| 22 | Imine | `[$([CX3]([#6])[#6]),$([CX3H][#6])]=[$([NX2][#6]),$([NX2H])]` |
| 23 | Isocyanate | `[NX2]=[C]=[O]` |
| 24 | Isothiocyanate | `[NX2]=[C]=[S]` |
| 25 | Ketone | `[#6][CX3](=O)[#6]` |
| 26 | Nitrile | `[NX1]#[CX2]` |
| 27 | Phenol | `[OX2H][cX3]:[c]` |
| 28 | Phosphine | `[PX3]` |
| 29 | Sulfide | `[#16X2H0]` |
| 30 | Sulfonamide | `[#16X4]([NX3])(=[OX1])(=[OX1])[#6]` |
| 31 | Sulfonate | `[#16X4](=[OX1])(=[OX1])([#6])[OX2H0]` |
| 32 | Sulfone | `[#16X4](=[OX1])(=[OX1])([#6])[#6]` |
| 33 | Sulfonic acid | `[#16X4](=[OX1])(=[OX1])([#6])[OX2H]` |
| 34 | Sulfoxide | `[#16X3]=[OX1]` |
| 35 | Thial | `[CX3H1](=S)[#6,H]` |
| 36 | Thioamide | `[NX3][CX3]=[SX1]` |
| 37 | Thiol | `[#16X2H]` |

## I    IMPACT OF DATA AUGMENTATION

Real-world experimental spectra inevitably exhibit variations due to instrumental noise and calibration drift. These issues are further aggravated by the limited availability of large-scale multi-modal spectral datasets. Since deep learning models require abundant data to generalize effectively, data augmentation serves as a key strategy. By artificially expanding the training set to simulate diverse experimental conditions, we can substantially improve the robustness, generalization ability, and predictive accuracy of the models. All experiments in this study were performed using molecular data sourced from the SDBS database.

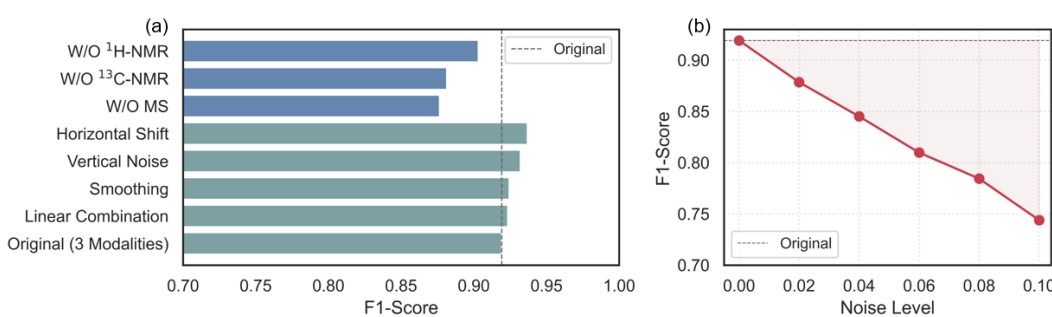

Figure 5: Evaluation of model robustness. (a) F1-scores under different data augmentation conditions and in modality-ablation studies. The dashed line indicates the performance of the model trained on original, complete data. (b) Impact of increasing levels of Gaussian noise on the final F1-score.

To this end, we explored several augmentation techniques. A random horizontal shift was applied to spectral signals to mimic peak displacement, while vertical perturbations were introduced by adding uniform random noise inversely scaled to the original signal intensity. In addition, Gaussian smoothing with a randomly selected bandwidth was employed to emulate instrument-dependent resolution effects. Beyond these perturbation-based methods, we also implemented a linear combination strategy, where two spectra from the same molecular class were blended with randomly assigned weights. This method effectively encourages the model to learn intermediate representations and smooth decision boundaries, while still preserving the chemical validity of the spectra.

Figure 5(a) presents the performance comparison. All augmentation strategies provided clear improvements over the baseline model trained without augmentation (F1-score = 0.9192). The most pronounced gain came from the **Horizontal Shift** augmentation, yielding an F1-score of 0.9369, highlighting the importance of accounting for peak misalignments. Vertical noise also contributed positively (0.9321), followed by Gaussian smoothing (0.9244). The linear combination approach achieved enhanced performance (0.9234), indicating that interpolating between spectra is a viable means of enriching the feature space. Overall, while every augmentation method enhanced predictive performance, the results suggest that carefully designed perturbations targeting realistic spectral variability are most effective in boosting model robustness.

## J    MODEL ROBUSTNESS UNDER IMPERFECT DATA CONDITIONS

To assess the model's practical utility, we conducted a series of experiments to probe its resilience against two common challenges in real-world spectral analysis: incomplete data and the presence of noise. The results, summarized in Figure 5, underscore the model's stability and ability to deliver reliable predictions even when faced with suboptimal data.

Our investigation into data completeness, shown in Figure 5(a), involved systematically withholding each of the three spectral modalities during testing. With the complete three-modality data, the model established a strong baseline F1-score of 0.9192. When a single modality was withheld to simulate incomplete data, the performance, while reduced, did not collapse. Specifically, the F1-score dropped to 0.9031 without $^1$H-NMR, 0.8814 without $^{13}$C-NMR, and 0.8765 without MS. This demonstrates that the model can effectively leverage the remaining available data to maintain a high level of predictive accuracy, a crucial feature for applications where acquiring a full suite of spectra is not always feasible.

Furthermore, we evaluated the model's tolerance to noise by injecting synthetic Gaussian noise of increasing intensity into the input spectra. This test mimics the random fluctuations inherent in experimental data. As illustrated in Figure 5(b), the model exhibited a smooth and predictable degradation in performance, with the F1-score declining steadily from 0.9192 on clean data to 0.7439 at the highest noise level of 0.1. Crucially, there was no sharp drop-off; this graceful degradation highlights the model's capacity to distinguish meaningful spectral features from random interference,

confirming its robustness for deployment in real-world analytical workflows where data quality may vary.

## K  SCAFFOLD-BASED DATA SPLIT

To prevent scaffold-level data leakage and create a rigorous test of generalization, we adopted an advanced clustering-based partitioning strategy. We first extracted Murcko scaffolds for all molecules and computed their Morgan fingerprints. Based on Tanimoto distances, these scaffolds were grouped into structurally homogeneous clusters via agglomerative clustering. To ensure a challenging split, we then employed a greedy algorithm that explicitly maximizes the structural dissimilarity between the training and test sets. This algorithm calculates the distances between all scaffold clusters and strategically allocates entire clusters to the test set to maximize its separation from the chemical space occupied by the training set.

The outcome of this diversity-maximization strategy is visualized in the t-SNE plot presented in Figure 6. The plot reveals a significant distributional shift between the training set (blue) and the test set (orange). The clear separation of their respective data clouds—highlighted by the distinct density contours—illustrates the structural disparity enforced by our splitting strategy. This partitioning provides a stringent evaluation setting, as the test molecules are structurally disjoint from the training set, compelling the model to generalize far beyond memorized structural patterns.

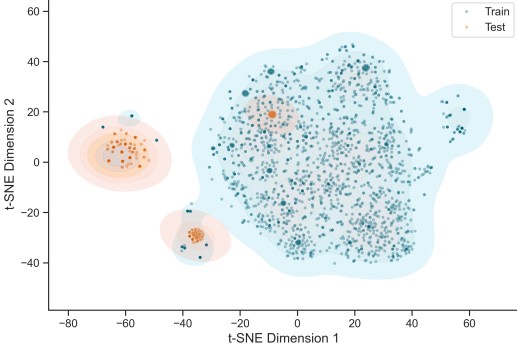

Figure 6: t-SNE visualization of the scaffold-based data split using our diversity maximization strategy. The training set (blue) and test set (orange) occupy separate regions in chemical space, demonstrating the effectiveness of the split.

## L  DECISION OF PREDICTING ORDER

The order in which functional groups are predicted can significantly influence the overall performance of multi-label prediction tasks. In Section 3.2, we systematically investigate the impact of prediction order by considering three distinct strategies: alphabetical, mutual information-based, and graph neural network (GNN)-based orderings. In this section, we provide a comprehensive description of the mutual information-based and GNN-based methods, emphasizing the intuition, construction steps, and practical implementation details behind each approach.

### L.1  MUTUAL INFORMATION-BASED ORDERING

This method leverages information-theoretic principles to infer the optimal prediction sequence among functional groups. The intuition is that functional groups with lower inherent uncertainty (entropy) are generally easier to predict, while groups exhibiting high dependency with others can be predicted more accurately when their correlated groups are known.

### L.1.1 INFORMATION-THEORETIC METRICS

For each functional group $f_i$, we quantify its uncertainty using entropy, a standard measure in information theory:

$$H(f_i) = -p(f_i = 1) \log_2 p(f_i = 1) - p(f_i = 0) \log_2 p(f_i = 0), \tag{39}$$

where $p(f_i = 1)$ and $p(f_i = 0)$ denote the probabilities of presence or absence of $f_i$ in the dataset, respectively. Intuitively, a functional group with lower entropy is more predictable, as its distribution is more concentrated.

To further understand dependencies between pairs of functional groups, we compute the conditional entropy, which captures the remaining uncertainty of $f_i$ given the state of $f_j$:

$$H(f_i|f_j) = \sum_{v \in \{0,1\}} p(f_j = v) \cdot H(f_i|f_j = v), \tag{40}$$

where

$$H(f_i|f_j = v) = -\sum_{u \in \{0,1\}} p(f_i = u|f_j = v) \log_2 p(f_i = u|f_j = v). \tag{41}$$

This allows us to measure how knowledge of one group reduces the uncertainty of another.

Based on these quantities, the mutual information between $f_i$ and $f_j$ is defined as:

$$I(f_i; f_j) = H(f_i) - H(f_i|f_j). \tag{42}$$

A higher mutual information value indicates a stronger dependency, meaning that knowing $f_j$ substantially reduces uncertainty about $f_i$.

### L.1.2 DEPENDENCY GRAPH CONSTRUCTION

After quantifying all pairwise dependencies via mutual information, we represent these relationships as a directed dependency graph $G = (V, E)$:

- Each node $v_i \in V$ represents a specific functional group $f_i$.
- A directed edge $(v_i, v_j) \in E$ is established if $f_i$ provides significant information for predicting $f_j$, as indicated by a mutual information value $I(f_i; f_j)$ exceeding a threshold $\tau$.
- Each edge is weighted by the mutual information value, reflecting the strength of the dependency.

This graph encodes the core relationships among functional groups, ensuring that the most informative dependencies are retained.

### L.1.3 MINIMUM SPANNING TREE CONSTRUCTION

To efficiently capture the most critical dependencies while avoiding redundancy, we transform the dependency graph into a distance matrix $D$:

$$D_{i,j} = \frac{1}{I(f_i; f_j) + \epsilon}, \tag{43}$$

where $\epsilon$ is a small constant to prevent division by zero. Using Kruskal's algorithm, we compute the minimum spanning tree (MST) over this graph:

$$\text{MST} = \arg\min_{T \subseteq G} \sum_{(u,v) \in T} D_{u,v}. \tag{44}$$

The resulting MST preserves the strongest dependencies and provides a tree structure that connects all functional groups with minimal total distance (i.e., maximal cumulative dependency).

## L.2    TRAVERSAL STRATEGY

To determine the prediction order, we initiate the traversal from the node corresponding to the most predictable functional group—that is, the node $v_s$ with the lowest entropy:

$$v_s = \underset{v_i \in V}{\arg\min} H(f_i). \tag{45}$$

We then traverse the MST using a breadth-first search (BFS) approach. At each step, the unvisited neighbors of the current node are sorted by their entropy in ascending order and appended to the queue. This process continues until all nodes have been visited. The resulting sequence prioritizes functional groups that are more predictable and respects the dependency structure encoded in the MST, thus facilitating more accurate sequential predictions.

## L.3    ORDERING VIA GRAPH NEURAL NETWORKS

We introduce a GNN-based framework to learn an interpretable and data-driven prediction order by modeling complex co-occurrence patterns and latent dependencies among functional groups.

**Co-occurrence Graph Construction.**    Consider a binary label matrix $\mathbf{Y} \in \mathbb{R}^{N \times K}$, where $N$ is the number of molecules and $K$ the number of functional groups. For each group $i$, we estimate its marginal probability:

$$p_i = \frac{1}{N} \sum_{n=1}^{N} Y_{ni}. \tag{46}$$

To capture pairwise dependencies, we compute the phi coefficient for each pair $(i, j)$:

$$\phi_{ij} = \frac{\mathbb{E}[Y_{:i} \wedge Y_{:j}] - p_i p_j}{\sqrt{p_i(1 - p_i)p_j(1 - p_j)}}, \tag{47}$$

where $\mathbb{E}[Y_{:i} \wedge Y_{:j}]$ is the empirical co-occurrence probability. An undirected, weighted graph $\mathcal{G} = (\mathcal{V}, \mathcal{E})$ is constructed, with nodes corresponding to functional groups and edges $(i, j)$ included if $\phi_{ij} > \tau$. Edge weights are set to $\phi_{ij}$, representing the strength of co-occurrence.

**Learning Functional Dependencies via GNN.**    To model higher-order dependencies, we encode each node using feature vectors $\mathbf{X} \in \mathbb{R}^{K \times K}$ (e.g., one-hot encodings). The graph structure (adjacency and edge weights) is fed into a GNN comprising GCN and GAT layers:

$$\mathbf{H}^{(1)} = \text{ReLU}(\text{GCNConv}(\mathbf{X}, \mathbf{A}, \mathbf{w})), \tag{48}$$

$$\mathbf{H}^{(2)} = \text{ReLU}(\text{GATConv}(\mathbf{H}^{(1)}, \mathbf{A})), \tag{49}$$

$$\mathbf{Z} = \text{GCNConv}(\mathbf{H}^{(2)}, \mathbf{A}), \tag{50}$$

where $\mathbf{A}$ is the adjacency matrix and $\mathbf{w}$ are the edge weights. Each node embedding $\mathbf{z}_i$ encodes the structural and dependency information for group $i$. The importance of node $i$ is then scored as:

$$s_i = \mathbf{w}_{\text{imp}}^{\top} \mathbf{z}_i + b_{\text{imp}}, \tag{51}$$

where $\mathbf{w}_{\text{imp}}$ and $b_{\text{imp}}$ are learnable parameters.

**Dependency-Aware Order Induction.**    A directed dependency graph $\mathcal{G}_{\text{dep}}$ is constructed using the learned embeddings and scores. For each pair $(i, j)$, a directed edge from $i$ to $j$ is added if $s_i < s_j$ and either the cosine similarity between embeddings exceeds a threshold $\delta$ or the phi coefficient is above $\tau$:

$$s_i < s_j \quad \text{and} \quad (\cos(\mathbf{z}_i, \mathbf{z}_j) > \delta \quad \text{or} \quad \phi_{ij} > \tau). \tag{52}$$

Edge weights are computed as a convex combination of semantic similarity and co-occurrence strength:

$$w_{ij} = \alpha \cdot \cos(\mathbf{z}_i, \mathbf{z}_j) + (1 - \alpha) \cdot \phi_{ij}, \tag{53}$$

with $\alpha \in [0, 1]$ (default $\alpha = 0.7$). To determine a global ranking, we apply the PageRank algorithm **?** to obtain a score $r_i$ for each node. The final ordering score is then:

$$o_i = \lambda s_i + (1 - \lambda)r_i, \tag{54}$$

where $\lambda \in [0, 1]$ (typically $\lambda = 0.6$). Functional groups are sorted in ascending order of $o_i$, yielding the final prediction sequence.

In summary, both the mutual information-based and GNN-based strategies provide principled frameworks for determining an effective prediction order by explicitly modeling dependencies and predictability among functional groups, thereby facilitating improved performance in sequential prediction tasks.

## M  MODALITY IMPORTANCE ON DIFFERENT SAMPLES

To analyze how the model assigns importance across spectra for samples of varying difficulty, we evaluated 500 test samples and recorded modality-level importance scores. Based on functional group counts, we identified the five easiest and hardest groups to predict. For these subsets, we visualized importance score distributions across input spectra (Figure 7). In the visualization, warmer regions indicate higher importance, i.e., where the model retains more information.

The importance maps reveal that the model tends to retain less information for easier samples, effectively discarding redundancy to enhance generalization. In contrast, for harder samples, it preserves more information to enable fine-grained analysis. Furthermore, the importance patterns vary significantly across samples, likely due to differences in spectral characteristics and the complementary nature of the modalities.

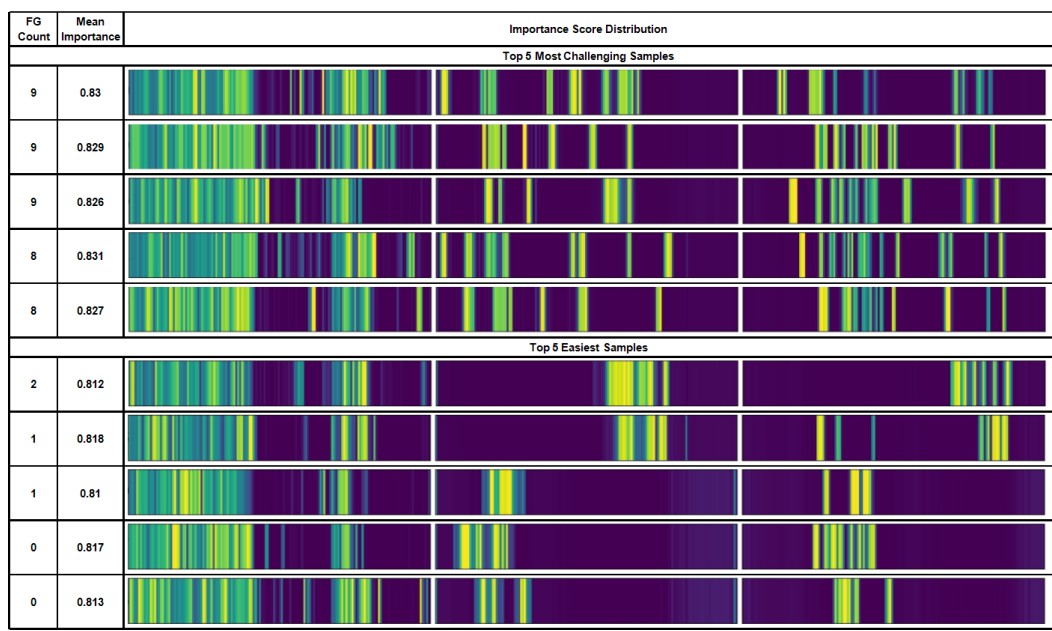

Figure 7: Importance maps for the top 5 most challenging and top 5 easiest samples. The first number denotes the number of functional groups in each molecule, and the second indicates the average importance score. From left to right: IR spectrum, $^{1}$H-NMR, and $^{13}$C-NMR. Warmer regions indicate higher importance, while cooler regions represent lower importance.

## N  VISUALIZATION OF THE DYNAMIC WEIGHTING STRATEGY

To further illustrate and validate the efficacy of our dynamic weighting strategy in mitigating data imbalance, we provide two visualizations. This strategy, based on Conditional Mutual Information (CMI), is designed to adaptively assign higher weights to more challenging samples and functional groups, thereby optimizing the model's learning process.

Figure 8 demonstrates the relationship between the average sample weight assigned to each molecule and the number of functional groups it contains. A clear positive correlation is observed: as the number of functional groups in a molecule increases, so does the average sample weight assigned by the model. This indicates that our model successfully identifies structurally complex molecules as more challenging examples and allocates greater attention to them during training. This aligns with our design objective of assigning higher importance to samples with greater prediction uncertainty.

Figure 9 offers a more granular view, showing how dynamic weights are adjusted based on the prediction difficulty of individual functional groups. In this plot, functional groups are sorted by their F1 score from lowest to highest. The results show that functional groups that are harder to predict (i.e., have lower F1 scores), such as Thial, Azo compound, and Phosphine, consistently receive higher dynamic weights. Conversely, high-frequency functional groups where the model performs well are assigned comparatively lower weights. This trend confirms that our dynamic weighting mechanism effectively identifies and amplifies the signal from underrepresented or hard-to-distinguish classes, preventing them from being overlooked during training.

Collectively, these two figures provide empirical evidence for our CMI-driven dynamic weighting strategy. They demonstrate its ability to steer the model's focus toward both difficult samples and difficult classes, which is crucial for achieving robust performance on imbalanced data.

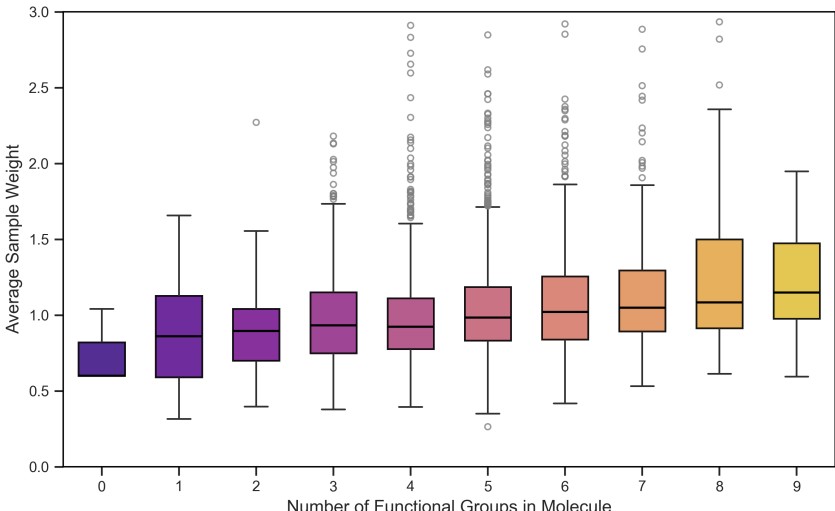

Figure 8: The relationship between the number of functional groups in a molecule and the average sample weight assigned. Molecules with more functional groups are considered more complex and are assigned higher weights.

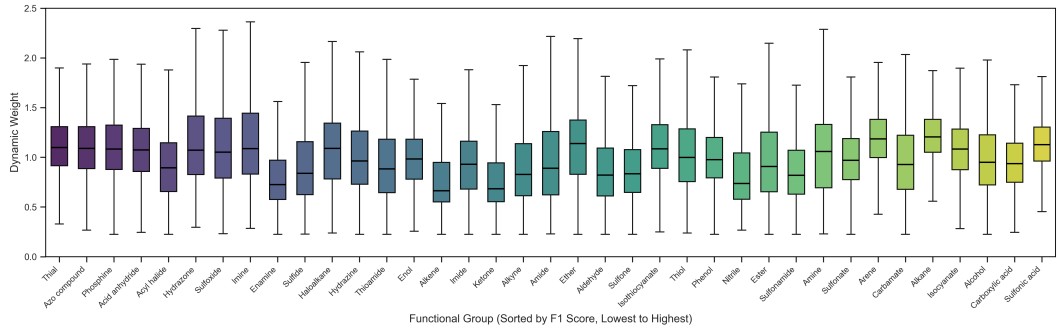

Figure 9: Dynamic weights assigned to functional groups. Lower-performing functional groups receive higher weights, focusing the model's attention on challenging classes.

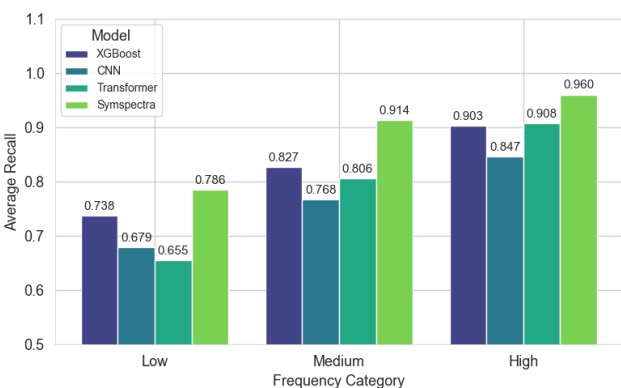

Figure 10: Average recall for functional groups grouped by frequency: low ($\leq$ 600), medium (601–5000), and high ($\geq$ 5000). Our method consistently outperforms others across all groups, particularly in the low-frequency regime, indicating improved robustness to label imbalance.

## O   FURTHER INVESTIGATION INTO LABEL IMBALANCE

To further quantitatively assess our model's adaptability to varying numbers of functional groups, we grouped the functional classes based on their occurrence frequencies[5]: low-frequency, medium-frequency, and high-frequency. For each group, we computed the average recall. As shown in Figure 10, our model achieves higher average recall than the baseline in all three categories. The performance gap is especially significant in the low-frequency group, where data scarcity typically limits learning. This result underscores the model's capacity to mitigate class imbalance by preserving discriminative features even for underrepresented functional groups.

## P   THE IMPACTION OF DECODER TYPE

The choice of decoder architecture significantly influences how the model utilizes the predictions of previously predicted functional groups. We experimented with GRU-based, LSTM-based, and Transformer-based decoders, with results summarized in Table 9. The LSTM-based decoder achieved a slight performance improvement of approximately 1% over the other methods. This could be attributed to its ability to balance complexity and sequential dependency modeling, while maintaining robustness against overfitting in smaller datasets. As a result, we selected the LSTM-based decoder as the final architecture for our model.

Table 9: Performance on various decoder.

| Decoder Type | F1 Score | Accuracy |
|---|---|---|
| LSTM | 0.965 | 0.7588 |
| GRU | 0.957 | 0.6988 |
| Transformer | 0.961 | 0.7165 |

## Q   HYPERPARAMETER EXPERIMENTS

To assess the impact of hyperparameter choices on model performance, we conduct a series of experiments by varying the information bottleneck trade-off coefficients $\beta$ from $10^{-8}$ to $10^{-3}$ and the sample emphasis $s_1$ and $s_2$ from 0.1 to 0.6. As shown in Figure 11, optimal performance occurs at $s_1 = s_2 = 0.3$ with $\beta = 1e - 6$.

---

[5]Grouping criteria: **Low frequency** ($\leq$600 samples): 14 groups. **Medium frequency** ($>$600 and $\leq$5000 samples): 9 groups. **High frequency** ($>$5000 samples): 14 groups.

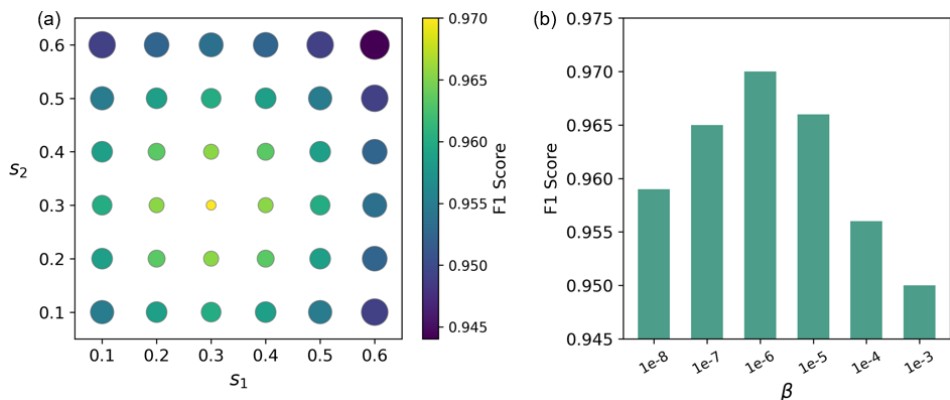

Figure 11: Hyperparameter Analysis. (a) Effect of the sample emphasis regulators $s_1$ and $s_2$ with circle size indicating error magnitude. (b) Impact of the information bottleneck coefficient $\beta$.

Table 10: Comparison of resource usage and performance against baseline models.

| Model | Mem. (GB) | Time (h) | F1-score |
|---|---|---|---|
| XGBoost | 209 | 0.7 | 0.913 |
| 1D-CNN | 5.7 | 2.0 | 0.900 |
| Transformer | 1.7 | 35.0 | 0.911 |
| **SymSpectra** | **7.0** | **2.7** | **0.965** |

Table 11: Performance and resource usage as a function of the number of modalities.

| #Modalities | Mem. (GB) | Time (h) | F1-score |
|---|---|---|---|
| 1 | 3.5 | 2.0 | 0.914 |
| 2 | 5.3 | 2.4 | 0.948 |
| **3** | **7.0** | **2.7** | **0.970** |
| 4 | 9.8 | 3.9 | 0.971 |
| 5 | 12.1 | 5.0 | 0.973 |

# R  COMPUTATIONAL COST AND MODALITY SELECTION

To assess the practical viability of our proposed model, SymSpectra, we conducted a comprehensive analysis of its computational resource requirements. We benchmarked its performance against several established baseline models and investigated the trade-off between predictive accuracy and computational cost as a function of the number of input modalities. The results of this analysis are summarized in Table 10 and Table 11.

Our analysis in Table 10 demonstrates that SymSpectra achieves a state-of-the-art F1-score of 0.965, significantly outperforming the baselines. Notably, it accomplishes this with a substantially lower training time (2.7 hours) compared to the Transformer model (35 hours) while maintaining a manageable memory cost of 7 GB.

The selection of three modalities is a deliberate decision aimed at optimizing the balance between performance and computational overhead. As detailed in Table 11, increasing the number of modalities from one to three yields a substantial performance gain, boosting the F1-score from 0.914 to 0.970 with only a moderate increase in resource utilization. However, further increasing the modalities to four or five, while offering marginal F1-score improvements, results in a disproportionate surge in resource consumption. For instance, moving from three to five modalities increases the memory footprint by 73% and training time by 85%, for only a minor improvement in the F1-score. This trade-off is particularly critical given that the acquisition of comprehensive, multi-modal spectral data is often infeasible in real-world applications, which strongly favors a more data-efficient model. Therefore, the three-modality configuration represents the most compelling trade-off, delivering near-peak performance while ensuring both computational efficiency and practical relevance.

