# OpenReview forum: "SymSpectra: Symmetric Information Bottleneck Framework for Molecular Structure Recognition under Imbalanced Settings"
_ICLR.cc/2026/Conference — ICLR 2026 Conference Withdrawn Submission_

### Official Review · Reviewer_QZ5c · 2025-10-23

**Soundness:** 2
**Presentation:** 3
**Contribution:** 3
**Rating:** 4
**Confidence:** 3

**Summary:**

The paper introduces SymSpectra, a technique for mitigating class imbalance in predicting the presence of functional groups from spectral data such as IR and NMR. The method is grounded in an information-theoretic formulation and is shown to outperform existing methods in the domain.

**Strengths:**

- The theoretical framework is innovative. To the best of my knowledge, a similar information-theoretic approach has not been explored for spectrometric or spectroscopic modalities before.
- The proposed method outperforms recently published studies.
- The work is comprehensive, with attention to detail from data preparation to ablations.
- The paper is accompanied by clean source code.

**Weaknesses:**

- My major concern is that the paper does not compare SymSpectra against standard class-imbalance techniques such as under/oversampling, inverse class-frequency loss weighting, focal loss, etc.
- Quite many details are unclear:
  - How were the baselines retrained?
  - How is the graph in Figure 1c constructed? Is it based on the first section of Appendix L.3?
  - Figure 2 is difficult to understand without a detailed caption; some terms in the figure do not seem to match the main text.
  - Equation 1: please specify what the modalities are and define I (mutual information).
  - Line 169: what are B, L, C?
  - Line 186: what exactly is H_j^I?
  - Equation 8: why does the training objective not involve labels Y?
  - Lines 318–320: the definitions of micro- and macro-averages appear to be swapped.
  - Line 347: please define “sample-level”.
  - Line 365: what are “importance distributions”?
  - Line 377: it is not clear what “shown” refers to
  - Line 398: how are the other splits constructed (e.g., those used in Figure 3a)?
  - Table 2: what do 8, 9, 13, 16, and 22 denote? Please name functional groups
- While it’s a minor point, it would be valuable to evaluate SymSpectra on a standard benchmark for spectra, such as MassSpecGym in the uni-modal setting (https://arxiv.org/abs/2410.23326).

**Questions:**

- Can the authors demonstrate that SymSpectra outperforms standard imbalance-mitigation techniques (e.g., class weighting, focal loss, under/oversampling) under the same training and evaluation protocol?

---

> ### Author Response · Authors · 2025-11-20
>
> We are grateful for the reviewer's helpful assessment of our manuscript. Below, we address each of your comments individually to provide further clarification and validation of our approach
>
> > W1&Q1: Comparison with Standard Imbalance Techniques**
>
> To rigorously demonstrate SymSpectra’s superiority, we conducted supplementary experiments enhancing the strongest baseline (Alberts et al.) and 1D-CNN with three established mitigation strategies: **Oversampling (MLSMOTE)**, **Inverse Class Frequency Reweighting**, and **Focal Loss**.
>
> | Model | Original | + Oversampling | + Reweighting | + Focal Loss |
> | :--- | :--- | :--- | :--- | :--- |
> | 1D-CNN | 0.900 | 0.910 | 0.915 | 0.917 |
> | Alberts et al. | 0.947 | 0.954 | 0.951 | 0.956 |
> | SymSpectra (Ours)| **0.970** | - | - | - |
>
> As shown above, while standard techniques improve baseline performance, SymSpectra (0.970) maintains a decisive advantage.
>
> This performance gap underscores a fundamental limitation of standard methods like Focal Loss, which apply **static, class-level** adjustments that treat all samples within a category equally.Unlike these heuristic baselines, SymSpectra’s CMI mechanism offers **dynamic, instance-level** guidance. By adaptively quantifying information uncertainty for *each specific sample*, it automatically prioritizes difficult instances regardless of their global class frequency, proving significantly more effective in identifying challenging substructures.
>
> ---
>
> > W2. Clarification of Details
>
> > 1. Baseline Retraining Details
>
> To ensure a comparison of model performance that is as fair as possible, we performed rigorous retraining for all baselines.
>
> * For 1D-CNN and Transformer: As these models originate from the paper by Alberts et al. (2024) [1] which introduced the multimodal dataset, they are directly applicable to the current multimodal setting. We used the architectures and parameters defined in that paper and retrained them using the multiple modalities.
> * For Wu et al. and Alberts et al.: These models were originally designed for IR spectra only (which is one of the main modalities in our dataset). We adapted their input modules to accept three types of spectra and performed **hyperparameter tuning** on key parameters to adapt them to the multimodal input.
>
> To improve clarity and reproducibility, we will include the implementations of these retrained baselines in the final version of our code.
>
> [1] Alberts M, Schilter O, Zipoli F, et al. Unraveling molecular structure: A multimodal spectroscopic dataset for chemistry. *Advances in Neural Information Processing Systems*, 2024, 37: 125780-125808.
>
> > 2. Construction of Figure 1c
>
> In Figure 1c, functional groups are treated as nodes. An undirected graph is constructed by connecting nodes with edges where the co-occurrence count exceeds a set threshold, using the co-occurrence count as the edge weight. We then utilized the Spring Layout algorithm to dynamically adjust node positions, causing tightly connected functional groups (high co-occurrence) to cluster together spatially. This graph was calculated using the large-scale dataset provided by Alberts et al. [1].
>
> **Clarification:** This process is distinct from the algorithm described in Appendix L.3. Figure 1c demonstrates **objective data statistics**, whereas L.3 describes the model's **subjective strategy** for determining the prediction order based on dependencies.
>
> > 3. Explanation of Figure 2 and Caption Update
>
> We acknowledge that Figure 2 requires a more detailed explanation. The workflow is as follows:
> 1.  The three spectra are first processed via linear interpolation to obtain fixed-dimension vectors, followed by a CNN to extract feature representations $X$.
> 2.  These multimodal features pass through the SCIB framework and a cross-modal attention mechanism to dynamically filter redundant information and extract key representations $T$. This process involves calculating the importance distribution of the spectra via an MLP and injecting noise, resulting in a KL divergence loss as part of the final optimization objective.
> 3.  Based on this, the model introduces a dynamic weighting strategy based on CMI. By comparing the outputs of the main predictor and auxiliary predictors (which use partial modal information), the model calculates sample-level weights ($W^{\text{samp}}$) and feature-level weights ($W^{\text{feat}}$).
> 4.  Finally, the LSTM decoder receives the fused features and generates the functional group prediction sequence combined with the prediction from the previous time step.

---

> > ### Author Response · Authors · 2025-11-20
> >
> > To avoid ambiguity, here we clarify all the terms used in the figure:
> > * $X_1, X_2, X_3$: Feature representations of the three spectra after CNN extraction. These correspond to Equation 2 in the paper. To avoid confusion with raw inputs, we will revise these labels to $O_1, O_2, O_3$.
> > * $T_1, T_2, T_3$: Intermediate representations obtained after spectral features are compressed via the Information Bottleneck. These correspond to Equation 6.
> > * $W^{\text{samp}}, W^{\text{feat}}$: Represent sample-level weights and feature-level weights, respectively, corresponding to Equations 11 and 12.
> > * $L_{assist1}$ is the prediction loss of the auxiliary predictor, $L_{pred}$ is the loss of the main predictor, and $L_{assist2}$ is the loss of the LSTM decoder.
> >
> > We will revise all the details in the final version.
> >
> > > 4. Definitions in Equation 1
> >
> > In Equation 1:
> > * **Modalities**: Refers to the spectral information from different modes; in our model, these are IR, $^{1}H$-NMR, and $^{13}C$-NMR.
> > * $I$: Represents **Conditional Mutual Information**. It quantifies the remaining dependency between variable $X$ and $Y$ given condition variable $Z$. In our model, this term represents the unique/incremental information that a specific modality can provide given the knowledge of other modalities.
> >
> > > 5. Definitions in Line 169 (B, L, C)
> > * $B$: **Batch size**, representing the number of molecular samples processed in one training iteration.
> > * $L$: **Sequence length**, representing the length of the spectra after linear interpolation. As stated in lines 838 and 851, this length is **600**, which is fully consistent with the description in the data preprocessing section.
> > * $C$: **Feature dimension**, specifically the hidden layer dimension after CNN encoding.
> >
> > > 6. Meaning of $H_j^i$ in Line 186
> >
> > $H_j^i$ represents the latent feature representation of the $j$-th Token in the $i$-th modality before entering the bottleneck layer. It represents the object being screened within the Information Bottleneck. The model determines whether to preserve this $H_j^i$ or replace it with Gaussian noise $\epsilon^i$ based on the importance score $\lambda_j^i$.
> >
> > > 7. Equation 8 and Label Y
> >
> > Following the Information Bottleneck principle, Equation 8 represents only the **compression objective** within the framework. It aims to compress redundancy as much as possible by minimizing the mutual information between the intermediate representation and the original information. This term leads to the calculation of the KL divergence loss mentioned later. It is not the total loss, and therefore does not include the label $Y$. The label $Y$ is involved in the **maximization of the prediction information term**, which corresponds to Equation 14.
> >
> > > 8. Definitions of Micro and Macro-F1
> >
> > For multi-label classification tasks, Micro-F1 and Macro-F1 are standard performance metrics. We have re-verified their definitions and confirmed they align with standards in machine learning literature:
> > * Macro-F1: Calculated as the arithmetic mean of per-class F1 scores. Because it treats every class equally (regardless of sample size), it is more sensitive to the performance on **Rare Classes**.
> > * Micro-F1: Calculated by aggregating TPs, FPs, and FNs globally. Therefore, it is dominated by **Frequent Classes**.
> >
> > To avoid ambiguity, we will add citations to relevant literature and refine the wording in the final version.
> >
> > [1] Farhadpour S, Warner T A, Maxwell A E. Selecting and interpreting multiclass loss and accuracy assessment metrics for classifications with class imbalance: Guidance and best practices. *Remote Sensing*, 2024, 16(3): 533.
> >
> > > 9. Definition of Sample-level
> >
> > As stated in lines 317-318: *"sample-level accuracy (ACC), the percentage of samples where all functional groups are correctly predicted."* For a given sample, its prediction is counted as correct only if **all** predicted functional group labels perfectly match the ground truth labels (i.e., Exact Match Accuracy).
> >
> > > 10. Meaning of Importance Distributions
> >
> > "Importance distributions" correspond to $p$ in **Equation 5**, which is the importance level of each spectral token within the entire spectrum. This directly corresponds to the retention probability of the spectral token in the SCIB module.
> >
> > > 11. Reference of "shown" in Line 377
> >
> > In Appendix M, we visualize the importance distributions of the three spectral modalities using Figure 7. The word "shown" refers to **"the samples to be shown"** in Figure 7. Specifically, we sorted the samples by functional group count and selected the top 5 and bottom 5; "shown" refers to these specific samples selected for visualization. We will revise the text in the final version to avoid ambiguity.

---

> > > ### Author Response · Authors · 2025-11-20
> > >
> > > > 12. Construction of Other Splits
> > >
> > > As described in line 321, we used an 8:1:1 random split for Figures 3a, 3b, and 3d, ensuring identical data across all models. In contrast, Figure 3c utilizes a scaffold-based split (see Appendix K). We clustered molecules via Murcko scaffolds and Morgan fingerprints, then partitioned them using a greedy algorithm. This ensures structural dissimilarity between sets, allowing us to measure the model's structural generalization.
> > >
> > > > 13. Meaning of Indices in Table 2
> > >
> > > The "Relative Sequence" indices in Table 2 correspond directly to the definition order of functional groups in **Table 5 of Appendix H**. The specific mapping is:
> > > * 8: Amide
> > > * 9: Amine
> > > * 13: Carboxylic acid
> > > * 16: Ester
> > > * 22: Imine
> > >
> > > To further avoid ambiguity, we will add an **index reference table** to the caption in the final version to improve readability.
> > >
> > > ---
> > >
> > > > W3. Regarding the MassSpecGym Dataset
> > >
> > > We appreciate the reviewer's suggestion. MassSpecGym is indeed a crucial and standard benchmark dataset in the field of mass spectrometry. However, we would like to clarify that the core research objective of this work is to explore leveraging the **Complementarity of Multi-modal Data** to mitigate the data imbalance problem arising from long-tail distributions.
> > >
> > > Specifically, our Symmetric Conditional Information Bottleneck (SCIB) framework is explicitly designed to extract complementary core information by facilitating information interaction and compression across different spectral modalities (IR, $^{13}C$-NMR, and $^{1}H$-NMR). In contrast, the MassSpecGym dataset focuses primarily on uni-modal mass spectrometry data. Evaluating our model on such a dataset would prevent the execution of cross-modal mutual information computation and dynamic weight assignment. Consequently, our core contribution—an imbalance learning strategy based on multi-modal fusion—could not be effectively validated.
> > >
> > > Furthermore, to demonstrate the robustness of our model architecture, we have presented evaluation results for SymSpectra under uni-modal input settings in **Table 1**. The results indicate that while our model remains competitive even in uni-modal settings, its true performance advantage stems from multi-modal fusion. Therefore, selecting a dataset containing multiple spectral types as the primary experimental platform aligns better with the design intent of this study.

---

> > > > ### Author Response · Authors · 2025-11-27
> > > >
> > > > Dear Reviewer,
> > > >
> > > > We sincerely thank you for your thoughtful insights provided in your review. We deeply appreciate the time and effort you dedicated to improving our work. We would be grateful if you could let us know whether our revisions and responses adequately address your concerns, or if there are any remaining points we can clarify.
> > > >
> > > > Best, The authors

---

### Official Review · Reviewer_Rok3 · 2025-10-29

**Soundness:** 2
**Presentation:** 3
**Contribution:** 2
**Rating:** 4
**Confidence:** 4

**Summary:**

This paper proposes a multimodal spectra fusion framework grounded in information theory, aiming to improve rare class functional group classification. The method fuses multiple spectroscopic modalities, including NMR, IR, MS, and MS/MS, and claims that the proposed model enhances cross-modal feature fusion by suppressing redundant information while retaining discriminative signals.
Experiments are conducted on two datasets, the NeurIPS 2024 benchmark dataset and the SDBS database. The authors report performance across various modality combinations.

**Strengths:**

The authors make an effort to ground the multi-modal fusion in information theory, attempting to provide a conceptual understanding of how IB may contribute to effective multi-modal feature selection.

The paper demonstrates reasonably strong performance across multiple modality combinations and datasets.

**Weaknesses:**

## Limited competing methods
The baseline selection appears weak and mostly borrowed from the NeurIPS 2024 benchmark models. The paper would benefit from including stronger and more recent competing methods in spectrum-based structure elucidation. If the goal is to demonstrate the advantage of multimodal fusion, it is essential to compare against the best single-modality SOTA models rather than relatively simple baselines. Recent relevant works such as DiffMS (ICML 2025), DiffSpectra (arXiv 2025), and LLM-based molecular reasoning benchmarks (NeurIPS 2024) should be considered or at least discussed.

## Gap between theory and implementation.
While the paper presents extensive information theory (IB, CMI), the implementation diverges from the stated formulation. In practice, the modality “information bottleneck” seems to be merely cross–attention–based modality weighting combined with element-wise feature summation and a shallow 1D CNN fusion.

CA weighting + summation + 1D CNN

Also, the proposed total loss function is closer to a focal loss style weighting, rather than a principled information-theoretic objective.

## Justification for improvements on rare functional groups.
The paper emphasizes improvements for rare groups, but the underlying cause of this improvement is not clearly attributed to a specific component of the method. Beyond empirical performance, a deeper analysis or ablation that isolates which module (SCIB, dynamic reweighting, sequential FG decoding, etc.) contributes to rare-class gains would strengthen the claim. At present, the explanation lacks theoretical or mechanistic grounding.

**Questions:**

See weakness

---

> ### Author Response · Authors · 2025-11-20
>
> We thank the reviewer for their rigorous assessment and valuable suggestions. We have taken your concerns seriously and offer the following detailed responses and additional evidence to address them.
>
> ---
>
> > W1. Limited Competing Methods
>
> **Clarification on Task Discrepancy**
>
> We appreciate the reviewer suggesting these recent works. However, we respectfully note that these models have fundamental task and setting differences that make them unsuitable as direct baselines for our specific task of **multi-label functional group classification**.
>
> * **DiffMS:** This model focuses on generating molecular graphs from mass spectrometry data using diffusion models(Bohde, 2025). Its objective is *de novo* generation rather than high-precision multi-label classification of substructures. Furthermore, DiffMS relies on SIRIUS for chemical formula assignment to embed MS peaks. SIRIUS utilizes combinatorial optimization based on fragmentation trees, a process specific to MS that does not generalize to other spectral modalities (IR, NMR) used in our framework.
> * **DiffSpectra:** This work focuses on the joint modeling of 2D and 3D molecular structures to predict molecular conformations from spectra (UV, IR, Raman). While 3D structure prediction is critical, it is a fundamentally different task from our functional group recognition. Moreover, DiffSpectra requires 3D conformational data for training, which is not present in our dataset.
> * **LLM-based Benchmarks:** While LLMs show potential, according to the paper(Guo, 2024), their current performance on fine-grained spectral analysis is significantly weaker than specialized models (F1-scores around ~0.6). The authors of the benchmark explicitly conclude that LLMs "struggle particularly in spectrum interpretation and molecule construction" due to their inability to process fine-grained spectral features. Therefore, they are not yet suitable state-of-the-art competitors for high-precision functional group prediction.
>
> To provide a comprehensive overview of the field, we will include a discussion of these works in the **Related Work** section of the final version.
>
> Bohde M, Manjrekar M, Wang R, et al. Diffms: Diffusion generation of molecules conditioned on mass spectra[J]. arXiv preprint arXiv:2502.09571, 2025.
> Guo K, Nan B, Zhou Y, et al. Can llms solve molecule puzzles? a multimodal benchmark for molecular structure elucidation[J]. Advances in Neural Information Processing Systems, 2024, 37: 134721-134746.
>
> **Strengthening Baselines with Advanced Imbalance Handling Techniques**
>
> To address the concern that our baselines might be "weak" regarding the severe class imbalance in this dataset, we extended our comparison by conducting a supplementary experiment to strictly validate SymSpectra’s superiority in handling data imbalance. We compared our method against established baselines (1D-CNN and Alberts et al. ) enhanced with three standard imbalance handling techniques: **Oversampling (MLSMOTE)**, **Inverse Frequency Class Reweighting**, and **Focal Loss**.
>
> | Model | Original | Oversampling (MLSMOTE) | Class Reweighting | Focal Loss |
> | :--- | :--- | :--- | :--- | :--- |
> | 1D-CNN | 0.900 | 0.910 | 0.915 | 0.917 |
> | Alberts et al. | 0.947 | 0.954 | 0.951 | 0.956 |
> | **SymSpectra (Ours)** | **0.970** | - | - | - |
>
> As shown above, SymSpectra (**0.970**) significantly outperforms strong baselines even when they are equipped with specialized imbalance handling techniques. Unlike heuristic strategies, our CMI-based mechanism dynamically prioritizes high-information samples without requiring external preprocessing.

---

> > ### Author Response · Authors · 2025-11-20
> >
> > > W2. Gap between Theory and Implementation
> >
> > We believe there is a misunderstanding regarding our architecture. The reviewer describes our SCIB module as simple "CA weighting + summation + 1D CNN" and our loss as "focal loss style weighting." We respectfully clarify that our implementation is a rigorous variational approximation of the Information Bottleneck (IB) objective, fundamentally different from deterministic weighting or focal loss in three key aspects:
> >
> > **1. Cross-Attention Estimates Conditional Mutual Information, Not Just Weights.**
> > In our framework, cross-modal attention is not a standard fusion layer. It is designed to estimate $p_{ij}$, the probability of preserving a specific spectral token $j$ *given* the other modalities (Eq. 3–5). By using one modality as the Query and the concatenation of others as Key/Value, we estimate the conditional importance of each token. A simple deterministic weighting (as assumed by the reviewer) would allow high-frequency patterns to dominate the attention map. Our approach specifically targets the term $I(Y; T^i | T^{\neg i})$, isolating unique information rather than redundant signals.
> >
> > **2. Stochastic Gating & Noise Channel for True Compression.**
> > Our compression is achieved through a **differentiable stochastic bottleneck**, not deterministic linear scaling. As shown in Eq. (6) and (7), we sample gates $\lambda_{ij}$ via the Gumbel-Softmax trick and combine them with Gaussian noise $\epsilon^i$:
> > $$T^i_j = \lambda^i_j H^i_j + (1-\lambda^i_j)\epsilon^i$$
> > This creates a randomized channel where irrelevant tokens are replaced by noise, thereby minimizing the upper bound of $I(X^i; T^i | X^{\neg i})$ (Eq. 17–19).
> > If this were simple "CA weighting" ($T^i = \sum \alpha H^i$), it would be a deterministic linear map—essentially a feature reordering that retains all information (including redundancy). Our ablation study (Table 3)  confirms this: removing the SCIB compression drops the F1-score for low-frequency groups from **0.866 to 0.827**, proving that the stochastic compression is vital for rare class performance.
> >
> > **3. CMI-based Dynamic Weighting $\neq$ Focal Loss.**
> > Focal Loss is a heuristic that reweights based on **prediction probability** ($p$). It amplifies gradients for "hard" samples (low $p$), regardless of whether the difficulty arises from informative rare features or simply noise/ambiguity in high-frequency classes.
> > In contrast, our dynamic weighting is governed by **Information Gain**. We estimate the Conditional Mutual Information (CMI) by measuring the performance drop of an auxiliary predictor under modality dropout (Eq. 9–12). We amplify samples and labels that possess **high information density** (high CMI), not necessarily those with low confidence. This ensures the model focuses on rare, structurally complex samples that provide unique information, rather than overfitting to noisy hard examples from dominant classes.
> >
> > In summary, the simplified architecture described by the reviewer corresponds to our ablated version (without stochastic gating and CMI reweighting), which we empirically proved to be inferior, particularly for imbalanced data.

---

> > > ### Author Response · Authors · 2025-11-20
> > >
> > > > W3. Justification for Improvements on Rare Functional Groups
> > >
> > > We have provided empirical evidence in our ablation study (Table 3) showing that removing specific modules degrades performance on rare groups. Here, we further clarify the mechanism behind these improvements:
> > >
> > > * **SCIB Module (Information Compression):** In **Appendix M (Figure 7)**, we visualize the importance score distribution. The model learns to retain *less* information for simple samples (filtering redundancy for generalization) and *more* information for complex/hard samples (preserving fine-grained details). This selective compression ensures that the model's representational capacity is not wasted on easy, high-frequency patterns, leaving sufficient capacity to encode the subtle signals of rare functional groups.
> > > * **CMI-based Dynamic Weighting (Bias Correction):** In **Appendix N (Figure 9)**, we analyze the correlation between assigned weights and functional group frequency. Rare groups like **Azo compounds** and **Phosphine** consistently receive significantly higher weights than common groups. This empirically proves that the CMI mechanism successfully identifies and amplifies the signal of underrepresented categories, preventing them from being "drowned out" by the gradients of dominant classes during optimization.
> > > * **Sequential FG Decoding (Dependency Modeling):** The sequential prediction explicitly models chemical dependencies (co-occurrence and mutual exclusion). For example, rare groups often appear in specific contexts alongside common "background" groups. By encoding these chemical priors via the LSTM decoder, the model leverages the presence of easy-to-predict groups to infer the presence of chemically related rare groups. This is evidenced by the significant performance boost in dependent groups like **Amides** and **Imines** when sequential decoding is enabled.
> > >
> > > We hope this clarification demonstrates that the improvements on rare classes are not accidental but are the direct result of our specific information-theoretic design choices.

---

> > > > ### Author Response · Authors · 2025-11-27
> > > >
> > > > Dear Reviewer,
> > > >
> > > > We sincerely thank you for your thoughtful insights provided in your review. We deeply appreciate the time and effort you dedicated to improving our work. We would be grateful if you could let us know whether our revisions and responses adequately address your concerns, or if there are any remaining points we can clarify.
> > > >
> > > > Best, The authors

---

### Official Review · Reviewer_mi8W · 2025-10-29

**Soundness:** 2
**Presentation:** 2
**Contribution:** 2
**Rating:** 4
**Confidence:** 3

**Summary:**

The paper introduces SymSpectra, a model designed to predict molecular substructures from multi-modal spectral data. While the proposed approach appears novel, its practical significance and methodological contribution are not fully clear.

**Strengths:**

- The paper addresses the problem of class imbalance, which is an important and underexplored challenge in machine learning for small molecules.
- The analyses of model performance in Section 3 and the Appendix are comprehensive.

**Weaknesses:**

Major comments

- Lack of clarity. The Abstract and Introduction emphasize class imbalance as the central motivation, yet Section 2 (Methodology) primarily focuses on multi-modality. These two issues appear to be intertwined without clear justification or structure. This weakens the overall narrative of the paper. Please also refer to the minor comments below.
- Practical relevance is unclear. It appears that Alberts et al. is the only non-trivial baseline (beyond simple 1D-CNN or Transformer) comparable to the proposed method (Wu et al. seems to be proposed for infrared spectra rather than multi-modal data). Furthermore, Section 3.1 notes that no publicly available multi-modal datasets beyond approximately 12K compounds exist. This raises concerns about the practical relevance and scalability of multi-modal spectral prediction. The paper should better justify why predicting molecular classes from multi-modal spectra is meaningful in real-world applications, supported by appropriate references.
- Unclear methodological novelty. The paper does not clearly explain how SymSpectra improves upon or differs from the approach of Alberts et al. A more explicit comparison of architectural components and methodological innovations is needed.
- Missing baselines for class imbalance. Although class imbalance is presented as the central challenge, the paper omits standard baseline techniques commonly used to address this issue, such as sample reweighting, undersampling/oversampling strategies [1], or focal loss for classification [2].

Minor Comments

- Figure 1:
  - Panel (a): Please specify the underlying dataset and clarify what the plotted values are, and their scale.
  - Panel (b): The comparison of significance across methods is unclear—shouldn’t significance be calculated for frequencies which is the main message of the panel?
  - Panel (c): The purpose and interpretation of this panel are unclear; please elaborate on its relevance.
- Figure 3d: The caption mentions “correlation,” but no correlation is analyzed in the figure itself. Please clarify or revise.

[1] He et al., 2009, Learning from Imbalanced Data. IEEE Transactions on Knowledge and Data Engineering

[2] Lin et al., 2017, Focal Loss for Dense Object Detection. arXiv:1708.02002

**Questions:**

1. What would be the performance (e.g., in Figure 3d) of the method proposed by Alberts et al. if combined with standard class imbalance mitigation techniques, such as class reweighting? (Please refer to the corresponding discussion in the Weaknesses section above.)

---

> ### Author Response · Authors · 2025-11-20
>
> We appreciate the reviewer’s insightful comments and the opportunity to clarify key aspects of our work. To better address your questions and resolve the issues raised, we have provided a comprehensive analysis below.
>
> ---
>
> > W1. Clarification of Narrative
>
> We acknowledge the reviewer's point regarding the connection between multi-modality and class imbalance. Here, we clarify the logic of our current narrative: Since our model is built upon multi-modal spectral analysis, multi-modal fusion is an indispensable foundational capability. Crucially, our core mechanisms—SCIB and CMI-based dynamic weighting—rely heavily on the properties of multi-modal data (e.g., utilizing robust multi-modal features to provide reliable information estimates for CMI reweighting). Consequently, multi-modal integration features prominently in the Methodology section. However, the descriptions of these core modules (SCIB, CMI, etc.) are expressly designed to address the **core challenge** of class imbalance, which is our primarily problem.
>
> Furthermore, we thank the reviewer for the minor comments and offer the following clarifications:
> * **Figure 1(a):** This statistic is derived from the dataset by Alberts et al. (containing ~794K samples) to illustrate the functional group class imbalance inherent in the data.
> * **Figure 1(b):** We clarify that the primary objective of this panel is to visualize the inherent difficulty gap across frequency tiers caused by data imbalance. The substantial performance degradation observed from 'High' to 'Low' frequency groups serves as clear empirical evidence of this challenge. Therefore, the figure is intended to qualitatively highlight the severity of the "long-tail" problem that motivates our work, rather than to perform formal significance testing between the prediction results of those models.
> * **Figure 1(c):** To demonstrate the necessity of capturing functional group dependencies, we visualize the co-occurrence relationships within the Alberts et al. dataset. In this graph, edges connect groups with co-occurrence frequencies above a threshold, and the proximity between nodes illustrates the strength of their association (where higher frequency corresponds to shorter distance).
> * **Figure 3(d):** This simultaneously displays the relationship between functional group frequency and Macro-F1 score, emphasizing the correlation between functional group frequency and model performance.
>
> We will incorporate these clarifications into the final version of the paper.
>
> ---
>
> > W2. Practical Value and Datasets
>
> **Baseline Selection:**
> Regarding baselines, the 1D-CNN and Transformer models we used are standard baselines established by Alberts et al. (2024); they are **optimized for this dataset** . Due to a lack of existing multi-modal baselines for this specific task, we adapted the originally single-modal model by Wu et al. (2025). By **modifying its input and optimizing parameters**, we achieved improved results (Table 1: F1-score 0.886 -> 0.944), validating the effectiveness of multi-modal fusion. Alberts et al. **represents the current state-of-the-art research** aiming to elucidate molecular structure from spectra and is the closest comparator to our task.
>
> Moreover, as shown in Table 1, our experiments cover **a comprehensive comparison from single-modal to multi-modal settings**; SymSpectra not only performs better in single-modal settings but also effectively demonstrates the gains from multi-modal fusion, maintaining a significant advantage on imbalanced data and rare functional groups.
>
> **Dataset Scale**
> First, we respectfully clarify that our evaluation is **not limited to the 12K dataset**. Instead, we adopted a two-pronged strategy: we utilized the large-scale Alberts et al. dataset (794K) to rigorously **verify the model's generalization capability**, while employing the smaller SDBS dataset (12K) to **demonstrate its practical value**. As shown in Table 1, the high performance on the large-scale dataset (F1=0.970) confirms the model's robust generalization, whereas the success on the specific experimental dataset (F1=0.919) proves its practical utility in real-world scenarios despite limited data size.

---

> > ### Author Response · Authors · 2025-11-20
> >
> > **Multi-modal Necessity**
> > Regarding the necessity and relevance of multi-modal spectra, as noted in the Introduction (lines 40–43), single-modal data often fails to capture comprehensive molecular information(Lee, 2025;Alberts et al., 2024). In fields like drug discovery, accurate analysis of complex molecules relies on the cross-validation of multi-modal information (Han et al., 2024).
> >
> >
> > [1] Han Z, Zhao J, Tang Y, et al. Machine learning integration of multi-modal analytical data for distinguishing abnormal botanical drugs and its application in Guhong injection[J]. Chinese Medicine, 2024, 19(1): 2.
> > [2] Lee G, Shim H, Cho J, et al. Machine-Learning Approach to Identify Organic Functional Groups from FT-IR and NMR Spectral Data[J]. ACS omega, 2025, 10(12): 12717-12723.
> > [3] Alberts M, Schilter O, Zipoli F, et al. Unraveling molecular structure: A multimodal spectroscopic dataset for chemistry[J]. Advances in Neural Information Processing Systems, 2024, 37: 125780-125808.
> >
> > ---
> >
> > > W3. Methodological Novelty
> >
> > We assert that SymSpectra presents a **fundamental architectural distinction** compared to Alberts et al., rather than a mere incremental improvement.
> > * Alberts et al. is based on a Transformer architecture, mining molecular structure by converting spectra into patches and performing **Sequence-to-Sequence translation**.
> > * In contrast, our model is based on CNN/MLP structures, utilizing **Information Bottleneck theory** to extract core information from spectral data to predict molecular substructures.
> >
> > Specifically, SymSpectra introduces three core mechanisms completely absent in Alberts et al. to address critical pain points in spectral analysis:
> > 1.  **Spectral Feature Extraction:** While Alberts et al. uses patch-based methods to preserve spectral details, this can accumulate noise. SymSpectra uses **SCIB** to dynamically suppress redundant information, which is critical for handling multi-modal redundancy and factors like spectral noise.
> > 2.  **Imbalance Handling:** Alberts et al. relies on data augmentation to mitigate imbalance. We innovatively incorporate **CMI** to propose a dynamic weighting mechanism that explicitly forces the model to focus on rare functional groups and high-uncertainty samples, solving the "long-tail" problem at the algorithmic level.
> > 3.  **Structure Dependency Modeling:** Alberts et al. implicitly learns the statistical laws of SMILES strings. Our model uses a **sequential decoder** to explicitly model the chemical dependencies of functional groups.
> >
> > Therefore, SymSpectra is a theory-driven framework that introduces significant structural and methodological innovations to solve redundancy and data imbalance in multi-modal spectra, distinguishing it fundamentally from Alberts et al. in both structure and algorithm.
> >
> > [1] Alberts M, Schilter O, Zipoli F, et al. Unraveling molecular structure: A multimodal spectroscopic dataset for chemistry[J]. Advances in Neural Information Processing Systems, 2024, 37: 125780-125808.
> > [2] Alberts M, Zipoli F, Laino T. Setting new benchmarks in AI-driven infrared structure elucidation[J]. Digital Discovery, 2025.
> > [3] Wu W, Leonardis A, Jiao J, et al. Transformer-Based Models for Predicting Molecular Structures from Infrared Spectra Using Patch-Based Self-Attention[J]. The Journal of Physical Chemistry A, 2025, 129(8): 2077-2085.
> >
> > > W4&Q1: Comparison with Standard Imbalance Techniques
> >
> > Following your suggestion, we rigorously validate the superiority of our approach, we conducted supplementary experiments benchmarking SymSpectra against standard class imbalance mitigation techniques. Specifically, we enhanced both the **1D-CNN** and **Alberts et al.** baselines using three established methods:
> > * **Oversampling:** Using **MLSMOTE** to synthesize samples for rare functional groups.
> > * **Class Reweighting:** Applying **Inverse Frequency Weighting** to penalize majority classes.
> > * **Focal Loss:** Utilizing focal loss to focus training on hard-to-classify examples.
> >
> > The comparative results (F1-scores) are presented below:
> >
> > | Model | Original | + Oversampling | + Class Reweighting | + Focal Loss |
> > | :--- | :--- | :--- | :--- | :--- |
> > | **1D-CNN** | 0.900 | 0.910 | 0.915 | 0.917 |
> > | **Alberts et al.** | 0.947 | 0.954 | 0.951 | 0.956 |
> > | **SymSpectra (Ours)**| **0.970** | - | - | - |
> >
> > As shown in the table, while standard techniques do improve baseline performance, **SymSpectra still maintains a decisive performance advantage**.
> >
> > This performance gap highlights a fundamental limitation of traditional methods like Focal Loss or Reweighting, which apply **static, class-level** adjustments. In contrast, SymSpectra’s CMI mechanism provides **dynamic, instance-level** guidance. By adaptively quantifying the information uncertainty for *each specific sample*, it automatically upweights difficult instances regardless of their global class frequency, proving significantly more effective than static approaches for identifying challenging substructures.

---

> > > ### Comment · Reviewer_mi8W · 2025-11-25
> > >
> > > I appreciate the author's responses.
> > >
> > > Could you please provide a statistical significance analysis of the results in the table above? Is the higher performance of SymSpectra statistically significant compared to the baselines?

---

> > > > ### Author Response · Authors · 2025-11-27
> > > >
> > > > According to your suggestion, we performed 8 independent experiments to benchmark SymSpectra against two models employing conventional strategies to mitigate class imbalance and calculated the corresponding p-values. The detailed results are summarized in the following table:
> > > >
> > > > | Model             | Training Strategy | F1-Score | p-value (vs. Ours) |
> > > > | --------------------- | --------------------- | ------------ | ---------------------- |
> > > > | SymSpectra (Ours) | Original              | 0.970    | -                  |
> > > > | Alberts et al.        | + Focal Loss          | 0.956        | $8.13 \times 10^{-3}$  |
> > > > | Alberts et al.        | + Oversampling        | 0.954        | $1.89 \times 10^{-3}$  |
> > > > | Alberts et al.        | + Class Reweighting   | 0.951        | $3.17 \times 10^{-3}$  |
> > > > | Alberts et al.        | Original              | 0.947        | $9.29 \times 10^{-4}$  |
> > > > | 1D-CNN                | + Focal Loss          | 0.917        | $6.82 \times 10^{-5}$  |
> > > > | 1D-CNN                | + Class Reweighting   | 0.915        | $2.41 \times 10^{-4}$  |
> > > > | 1D-CNN                | + Oversampling        | 0.910        | $3.09 \times 10^{-5}$  |
> > > > | 1D-CNN                | Original              | 0.900        | $8.15 \times 10^{-6}$  |
> > > >
> > > > The results demonstrate that SymSpectra achieves statistically significant superiority over the baselines, **proving its effectiveness in handling data imbalance compared to traditional resampling or loss reweighting strategies**.

---

### Official Review · Reviewer_6ufe · 2025-11-01

**Soundness:** 3
**Presentation:** 2
**Contribution:** 3
**Rating:** 4
**Confidence:** 3

**Summary:**

This paper presents SymSpectra, a framework designed to identify molecular functional groups from multi-modal spectroscopic data (IR, NMR, MS) while addressing class imbalance challenges. The core contribution is a Symmetric Conditional Information Bottleneck (SCIB) approach that fuses spectral modalities, combined with conditional mutual information (CMI)-based dynamic weighting to prioritize rare functional groups and an LSTM decoder to model functional group dependencies. The authors report achieving an F1-score of 0.970 on simulated data and demonstrate robustness under various imbalanced scenarios.

**Strengths:**

Well-motivated problem: The class imbalance issue in functional group prediction is clearly articulated with supporting evidence (Figure 1), showing a 38% F1-score gap between high and low-frequency groups.
Comprehensive experimental validation: The paper includes extensive experiments across multiple dimensions such as multiple baselines (1D-CNN, Transformer, recent SOTA methods), both simulated (794K molecules) and experimental (12K molecules from SDBS) datasets and multiple imbalance scenarios(label imbalance, structural imbalance)

Thoughtful architectural design: The SCIB framework provides a principled approach to multi-modal fusion by explicitly modeling what information each modality uniquely contributes (Equation 1), moving beyond simple concatenation or attention-based fusion.

Interpretability efforts: The paper provides visualization of importance weights (Figure 7) and demonstrates that the model adaptively allocates attention based on sample complexity (Figure 8, Appendix M).

**Weaknesses:**

1. Limited theoretical justification for SCIB design. The paper drops the I(Y; T^¬i) term from Equation 15 citing "optimization instability" (Section E.1.2), but provides no empirical evidence of this instability. This is a significant theoretical compromise that weakens the SCIB framework's foundational claim.

2. The Gaussian assumptions in Equations 18-19 and 24-25 are strong but not validated. Why should spectral representations follow Gaussian distributions? The paper acknowledges exploring other priors (Appendix G, Table 4) but doesn't explain why Gaussian is fundamentally appropriate for spectral data.

3. The connection between minimizing I(T^i; X^i | X^¬i) and the specific attention-weighted formulation (Equations 26, 30) involves several non-trivial steps that are relegated to the appendix without sufficient intuition in the main text.

4. CMI-based weighting lacks rigorous justification. The paper claims CMI "quantifies and amplifies the informational significance" of rare groups (lines 96-99), but the actual implementation uses auxiliary predictors with modality dropout as a proxy (Section 2.2.3). The connection between this dropout-based approximation and true CMI is not formally established.

5. Equations 11-12 introduce normalization with batch statistics (μ_s, σ_s, μ_f, σ_f) and scaling factors (s1, s2), which appear ad-hoc. Why is this specific normalization scheme information-theoretically justified? The sensitivity analysis (Appendix Q, Figure 11) shows performance is sensitive to these hyperparameters, suggesting they're tuning knobs rather than principled quantities.
6. The claim that this approach is superior to "traditional class reweighting or resampling heuristics" (line 98) is not empirically validated, there is no comparison with focal loss, class-balanced loss, or re-sampling methods is provided.

7. The 1D-CNN and Transformer baselines seem weak (Table 1). For example, the Transformer achieves only 0.881 F1 on IR spectra while SymSpectra gets 0.924. This gap is suspiciously large and raises questions about whether baselines were properly tuned. The paper provides no details on baseline hyperparameters or architecture specifics.

8. Functional group dependency modeling seems oversimplified. The LSTM decoder uses a "predefined order" based on IUPAC nomenclature (Section 3.4, Table 2), which is essentially a chemical convention, not a learned dependency structure. While the paper explores MI-based and GNN-based ordering (Appendix L), these are not used in the final model.

9. The sequential prediction assumes a linear chain dependency structure, but functional group co-occurrence patterns (Figure 1c) suggest a more complex graphical structure. Why not use a conditional random field (CRF) or graph-based structured prediction?

9. The ~12K SDBS experimental dataset is much smaller and shows lower absolute performance across all models (0.919 vs 0.970 F1). This gap raises concerns about sim-to-real transfer, yet the paper doesn't deeply investigate why this gap exists or propose methods to close it.

10. Table 7 shows SymSpectra uses 7GB memory and 2.7 hours training, but what hardware? What batch size? The comparison with Transformer (1.7GB, 35 hours) seems inconsistent, why would the simpler model use less memory but far more time?

**Questions:**

1. Can you provide empirical evidence of the "optimization instability" claimed for minimizing I(Y; T^¬i)? Have you tried alternative formulations or regularization techniques to include this term?
2. How accurate is the modality dropout-based approximation of CMI? Can you provide theoretical analysis or empirical validation comparing the estimated CMI values with ground truth (if computable on synthetic data)?
3. Comparison with class imbalance methods: Can you compare SymSpectra with established techniques for handling class imbalance (focal loss, cost-sensitive learning, SMOTE, etc.) to validate that the CMI-based approach is indeed superior?
4. Have you considered more sophisticated dependency structures beyond sequential LSTM? Can you compare with CRF or graph neural network-based structured prediction?
5. What causes the significant performance drop from simulated (0.970 F1) to experimental (0.919 F1) data? Can domain adaptation or transfer learning techniques help?
6. What happens if you remove the information bottleneck term entirely (β=0) and just use the prediction loss with CMI weighting? This would help isolate the contribution of SCIB vs. CMI.
7. Table 2 shows different orderings yield substantially different results (e.g., Imine: 0.756 for GNN order vs 0.885 for IUPAC). How sensitive is the final model to ordering choice, and does this suggest the LSTM is exploiting spurious sequential patterns rather than true dependencies?

---

> ### Author Response · Authors · 2025-11-20
>
> We sincerely thank the reviewer for the constructive feedback and the time spent evaluating our work. We have carefully considered your comments and provide a detailed point-by-point response below to address your concerns.
>
> ---
>
> > W1 & Q1: **Lack of empirical evidence for removing $I(Y; T^{\neg i})$ term**
>
> We appreciate the your valuable feedback on the theoretical justification for removing the $I(Y; T^{\neg i})$ term from the objective function. As pointed out, including this term would create a fundamental conflict with the main predictor's objective, which is to maximize the joint mutual information $I(Y; T^i, T^{\neg i})$. Specifically, minimizing $I(Y; T^{\neg i})$ would force the intermediate representation $T^{\neg i}$ to carry no information about $Y$, contradicting the goal of the main predictor.
>
> To provide empirical evidence for this claim, we conducted an experiment in which we incorporated the upper bound of $I(Y; T^{\neg i})$ into the training objective. This term was approximated by calculating the KL divergence between $p(T^{\neg i})$ and the variational distribution $r(Y)$, following the Variational Information Bottleneck (VIB) framework (Alemi et al., 2016). We set $r(Y)$ as a standard Gaussian distribution $N(0,1)$ and introduced a hyperparameter $\theta$ to control the weight of this term in the objective function.
>
> The results from this experiment are shown below:
>
> |  $\theta$ | 0     | 0.01  | 0.1   | 1.0   |
> | ------------------- | ----- | ----- | ----- | ----- |
> |    f1-score                 | 0.970 | 0.956 | 0.952 | 0.931 |
>
> As shown, as the weight of $I(Y; T^{\neg i})$ increases, the model performance deteriorates. The model achieves the best performance when this term is excluded entirely, which aligns with our design choice of not including this term in the final objective.
>
> ---
>
> > W2: **Validation of the Gaussian Assumption**
>
> Apart from the experiments in Appendix G, we would like to clarify that the choice of a Gaussian distribution for our representations is grounded in the Central Limit Theorem. Specifically, our representation is formed by aggregating a large number of local features, a process that includes pooling and attention mechanisms (refer to Equations 2, 3, 4, and 5). According to the Central Limit Theorem, when aggregating a large number of independent or weakly dependent random variables (in our case, the local features of spectrum), the resulting distribution naturally converges to a Gaussian distribution, regardless of the original distribution of the individual features.
>
> To empirically validate this assumption, we conducted tests on the feature distributions of the three spectral modalities (IR, 13C-NMR, and 1H-NMR). Specifically, we tested the compressed feature distributions for these spectra using the Shapiro-Wilk and Kolmogorov-Smirnov (K-S) tests for normality.
>
> |              | IR     | ${}^{13}$C-NMR | ${}^1$H-NMR |
> | ------------ | ------ | -------------- | ----------- |
> | Shapiro-Wilk | 0.1648 | 0.1199         | 0.1479      |
> | K-S          | 0.5610 | 0.4210         | 0.4715      |
>
> As shown in the results, the p-values from both tests suggest that the distributions of the compressed features for all three spectral types (IR, 13C-NMR, and 1H-NMR) align with the Gaussian distribution assumption.
>
> ---
>
> > W3: **Clarification of Equations 26 and 30**
>
> To improve clarity of Equations 26 and 30, we have already included a formal proof of these equations in Appendix E. However, to ensure the explanations are more intuitive for the main text, we commit to providing a more accessible explanation in the revised manuscript.

---

> > ### Author Response · Authors · 2025-11-20
> >
> > > W4$Q2: **Lack of Rigorous Proof for CMI Approximation**
> >
> > As outlined in Equation (9), we define the conditional mutual information (CMI) $I(Y; T^i \mid T^{\neg i})$ as
> >
> > $I(Y; T^i \mid T^{\neg i}) = \mathbb{E}_{T^i, Y \mid T^{\neg i}} \left[ \log \frac{p(Y \mid T^i, T^{\neg i})}{p(Y \mid T^{\neg i})} \right].$
> > In practice, we use two predictors to obtain a variational approximation of the two conditional distributions in the above expression:
> >
> > * $p(Y \mid T^i, T^{\neg i})$ captures the full predictive power when all modalities are available, and is implemented by the **main predictor** with all modalities present.
> > * $p(Y \mid T^{\neg i})$ captures the predictive power from $T^{\neg i}$ alone, and is implemented by the **auxiliary predictor** where modality $i$ is dropped.
> >
> > Directly estimating CMI in high-dimensional continuous representation spaces is well known to be statistically and computationally challenging(Paninski, 2003). Therefore, we follow a standard variational strategy: we construct two parametric conditional distributions via the main and auxiliary predictors, and use their performance gap as a stable proxy for the log-ratio term in the CMI definition.
> >
> > **Experimental evidence**
> >
> > As shown in Fig. 8 and Fig. 9 of the appendix, harder samples and low-frequency functional groups indeed receive systematically larger weights, providing empirical evidence that the CMI-based weighting behaves as intended rather than as an ad-hoc heuristic.
> >
> > To further illustrate how CMI sharpens the model’s focus on the most informative modalities, we report the change in accuracy (Δ Accuracy) per modality for five representative functional groups:
> >
> > | Functional Group | IR Importance (Δ Accuracy) | HNMR Importance (Δ Accuracy) | CNMR Importance (Δ Accuracy) |
> > | ---------------- | -------------------------- | ---------------------------- | ---------------------------- |
> > | Sulfide          | **+0.2065**                | -0.0067                      | -0.0073                      |
> > | Sulfonamide      | +0.0087                    | **+0.0103**                  | -0.0185                      |
> > | Phenol           | **+0.0295**                | -0.0215                      | -0.0204                      |
> > | Thiol            | +0.0168                    | +0.0108                      | **+0.0414**                  |
> > | Isothiocyanate   | -0.0176                    | -0.0195                      | **+0.0184**                  |
> >
> > For rare or challenging functional groups such as Sulfide, Phenol, Thiol, and Isothiocyanate, CMI-based weighting consistently amplifies the contribution of the most informative modality (IR for Sulfide/Phenol, $^{13}$C-NMR for Thiol/Isothiocyanate). This pattern is precisely what one would expect from the CMI definition and strongly supports our claim that the proposed approximation **quantifies and amplifies the informational significance** of rare functional groups in a principled manner.
> >
> > [1] Paninski, L. *Estimation of entropy and mutual information*. Neural Computation, 15(6):1191–1253, 2003.
> >
> > ---
> >
> > > W5: **Clarification of Design Principles**
> >
> > **batch statistics**
> >
> > In Equations 11-12, our goal is to turn CMI estimates into numerically stable loss weights while preserving their **relative information content**. The batch statistics ($\mu_s, \sigma_s, \mu_f, \sigma_f$) standardize CMI into z-scores, similar to batch normalization. This is not ad-hoc: the absolute scale of CMI depends on the variational estimator and parameterization, so what matters for weighting is how much a sample’s CMI **deviates from the batch average**, not its raw magnitude. The mapping
> >
> > $$W = \max\{0,1 + s \cdot \tfrac{\text{CMI} - \mu}{\sigma}\}$$
> > is thus a simple linear transform from standardized information gain to non-negative weights, keeping the CMI-induced ordering while preventing exploding weights.
> >
> > **scaling factors**
> >
> > The scaling factors $s_1, s_2$ control the trade-off between the base prediction loss and the CMI-driven emphasis. As shown in Appendix Q, **performance is stable across a reasonable range**, with only extreme values causing degradation, which is expected for any trade-off parameter. This suggests the method is robust and the normalization provides a principled way to convert CMI into well-scaled loss weights.

---

> > > ### Author Response · Authors · 2025-11-20
> > >
> > > > W6&Q3: Comparison of Class Imbalance Methods
> > >
> > > Following your advice, we have added a supplementary experiment to further explore SymSpectra's performance on imbalanced data by comparing it with standard class imbalance handling techniques. This comparison was conducted on the simulated dataset from *Alberts et al.*.
> > >
> > > Specifically, we trained two models with completely different architectures (1D-CNN and Alberts et al.) to comprehensively evaluate the impact. The techniques used were:
> > >
> > > * **Oversampling:** Using MLSMOTE to generate new samples containing rare functional groups.
> > > * **Class reweighting:** Assigning higher weights to rare classes, specifically using inverse frequency weighting.
> > > * **Focal loss:** A commonly used method for addressing class imbalance.
> > >
> > > The results of these experiments are as follows:
> > >
> > > | Model             | Original | Oversampling | Class Reweighting | Focal Loss |
> > > | ----------------- | -------- | ------------ | ----------------- | ---------- |
> > > | 1D-CNN            | 0.900    | 0.910        | 0.915             | 0.917      |
> > > | Alberts et al.    | 0.947    | 0.954        | 0.951             | 0.956      |
> > > | SymSpectra (Ours) | 0.970    | -            | -                 | -          |
> > >
> > > SymSpectra significantly outperforms both methods, achieving an F1 score of 0.970, which is notably higher than both 1D-CNN and Alberts et al. across all tested techniques. Demonstrating it's ability  to focus on  underrepresented classes without requiring additional preprocessing.
> > >
> > > ---
> > >
> > > > W7: Concern Regarding Baseline Strength
> > >
> > > Regarding the **1D-CNN** and **Transformer** baselines, these models were not arbitrarily selected for this work. Instead, we directly adopted the **official benchmark architectures and hyperparameter configurations** established by **Alberts et al. (2024)** upon the release of this multimodal spectroscopic dataset. To ensure **fairness and comparability**, we strictly reproduced the original authors' codebase and training strategies without introducing subjective modifications. This guarantees that our evaluation benchmarks align with the established standards in this domain. To ensure full transparency, we will include the complete reproduction scripts for these baseline models in our final code release.
> > >
> > > We contend that the performance gap observed in Table 1 stems not from inadequate tuning of the baselines, but rather from **inherent limitations within these architectures**:
> > > * CNNs may introduce significant redundancy when processing multimodal data, potentially obscuring discriminative features.
> > > * Standard Transformers tend to overfit to high-frequency head classes while underperforming on rare functional groups due to the severe data imbalance.
> > >
> > > In contrast, **SymSpectra** explicitly addresses these "pain points" by incorporating **Conditional Mutual Information (CMI)-based dynamic weighting** and the **Information Bottleneck (IB)** mechanism.
> > >
> > > ---
> > >
> > > > W8, W9 & Q4: Dependency Modeling & Ordering Strategy
> > >
> > > **1. LSTM as an Efficient Approximation of Graph Structures:**
> > > While we acknowledge that the dependencies between functional groups inherently form a graph structure, exact inference methods like CRFs often entail prohibitively high computational complexity for structured prediction. In contrast, the LSTM decoder represents a strategic design choice balancing **efficiency and performance**. As shown in **Table 2**, this approximation yields significant performance gains in predicting context-dependent groups (e.g., Aldehyde) compared to non-sequential CNN baselines, demonstrating that the module successfully captures critical chemical dependencies.
> > >
> > > **2. Rationale for IUPAC Ordering:**
> > > In this paper, we explored the impact of various ordering strategies, including the predefined **IUPAC** order and data-driven orders computed via **GNN** and **Mutual Information (MI)**. Although data-driven orders can capture statistical co-occurrences within the training set, they are prone to overfitting dataset-specific biases, leading to poor generalization in real-world scenarios. In contrast, the predefined IUPAC order serves as a **domain-invariant prior** consistent with established chemical rules. It does not rely on specific dataset distributions, thereby ensuring the model learns stable, chemically meaningful dependencies and enhancing its generalization capability.

---

> > > > ### Author Response · Authors · 2025-11-20
> > > >
> > > > > W10: Sim-to-Real Gap
> > > >
> > > > It is true that the performance on the SDBS dataset is lower than on the Alberts et al. simulated dataset. However, this gap is primarily attributed to the extreme reduction in data scale (from 794K down to ~12K samples) and realistic factors such as noise and peak shifts inherent in experimental spectra.
> > > >
> > > > Despite the reduction in training data, the model did not suffer from catastrophic failure but instead maintained a high F1-score. Furthermore, as detailed in **Appendix I**, we investigated the impact of various data augmentation techniques. We demonstrated that performance can be recovered simply by applying realistic perturbations (e.g., horizontal shifts), proving that this performance gap is not a defect of the model architecture but rather a domain shift that can be mitigated.
> > > >
> > > > > W11: Details on Computational Resources
> > > >
> > > > We apologize for the brevity of the initial description. As mentioned in **Appendix B**, all models were trained on a single **NVIDIA A100 (80GB) GPU**, with a batch size of 128 for SymSpectra.
> > > >
> > > > Regarding the training time discrepancy (**35h vs. 2.7h**): To ensure a fair comparison, we strictly reproduced the official settings from Alberts et al. (2024) for the Transformer baseline, utilizing a **batch size of 4096 tokens** and **training for 250,000 steps**. The Transformer architecture is inherently more difficult to converge on this multimodal task, necessitating significantly more iteration steps.
> > > >
> > > > > Q6: The Contribution of SCIB vs. CMI
> > > >
> > > > We conducted a rigorous ablation study in **Table 3**. The row labeled **"- SCIB Compression"** corresponds to the setting where the information bottleneck term is completely removed ($\beta=0$), retaining only the CMI-weighted prediction loss. The results are as follows:
> > > >
> > > > | Component | High | Medium | Low |
> > > > | :--- | :--- | :--- | :--- |
> > > > | **SymSpectra** | **0.967** | **0.942** | **0.866** |
> > > > | **- CMI** | 0.957 | 0.922 | 0.844 |
> > > > | **- SCIB ($\beta=0$)** | 0.949 | 0.919 | 0.827 |
> > > >
> > > > As shown in the table, removing either module negatively impacts the final prediction results. Furthermore, the exclusion of the SCIB module leads to a performance decline for functional groups of all frequencies. In contrast, removing CMI disproportionately affects low-frequency groups. This suggests that SCIB facilitates global representation learning by reducing redundancy, while CMI specifically boosts the model's ability to predict low-frequency targets.
> > > >
> > > >
> > > > > Q7: Sensitivity to Prediction Order
> > > >
> > > > In **Table 2**, we indeed show that the LSTM module is sensitive to the prediction order, and the performance for specific functional groups is closely related to their position in the sequence.
> > > >
> > > > However, we argue that this sensitivity is **not a model defect**, but rather evidence that the model is successfully capturing **true dependency relationships**. The model can only effectively utilize dependency information when the context is chemically valid (i.e., the correct prerequisite groups are predicted first). If the model were insensitive to the relative order of functional groups or showed random performance fluctuations, it would suggest that the model was exploiting spurious patterns or noise.
> > > >
> > > > Moreover, for the final model, although local predictions vary with order, the use of the predefined **IUPAC** order eliminates the influence of dataset-specific bias. We believe this ordering strategy contributes to better generalization by enforcing a consistent, chemically grounded dependency path.

---

> > > > > ### Author Response · Authors · 2025-11-27
> > > > >
> > > > > Dear Reviewer,
> > > > >
> > > > > We sincerely thank you for your thoughtful insights provided in your review. We deeply appreciate the time and effort you dedicated to improving our work. We would be grateful if you could let us know whether our revisions and responses adequately address your concerns, or if there are any remaining points we can clarify.
> > > > >
> > > > > Best, The authors

---

### Author Response · Authors · 2025-12-01
**Final Summary for Area Chair**

Dear Area Chair,

We sincerely thank you for your time and dedicated service. We are encouraged to note that all reviewers provided valuable and constructive feedback **without raising fundamental objections to the novelty or validity of our core methodology**. We understand that the initial conservative ratings stemmed **primarily from requests for stronger empirical benchmarks and theoretical clarifications**, rather than inherent flaws in the work.

During the rebuttal phase, we thoroughly addressed the raised concerns by conducting extensive supplementary experiments and providing the requested theoretical proofs. We believe these additions effectively resolve the reviewers' comments. For clarity, we summarize the key strengths and our responses below:

> **Consensus on Contributions:**

* **Innovative Theoretical Framework**: The SCIB framework is recognized as a principled and innovative approach which effectively models unique modality contributions beyond simple fusion. Reviewers highlighted that a similar **information-theoretic formulation has not been explored before**. (`Reviewers 6ufe, Rok3, QZ5c`)
* **Well-Motivated Problem Solving**: This work addresses the critical and underexplored challenge of class imbalance in spectral analysis with **clear articulation and strong empirical support**. (`Reviewers 6ufe, mi8W`)
* **Comprehensive Experimental Validation**: The evaluation is acknowledged as **extensive and rigorous**, demonstrating superior performance across diverse baselines, datasets (simulated & real), and multiple imbalance scenarios. (`All Reviewers`)
* **Interpretability and Reproducibility**: Reviewers appreciated the detailed visualization of importance weights for interpretability and the provision of clean source code for reproducibility. (`Reviewers 6ufe, QZ5c`)

> **Responses to Major Concerns:**

While commending the "**thoughtful architectural design**", `reviewer 6ufe` primarily focused on the theoretical grounding of our framework, requesting a rigorous justification for the multi-modal fusion mechanism. In our response, we provided rigorous empirical validations, including normality tests and ablation studies which substantiated the **validity of our theoretical assumptions**. Additionally, we provided detailed analyses confirming that **the multi-modal fusion mechanism behaves as theoretically intended**, effectively identifying and amplifying informative signals for rare groups. We believe these detailed additions fully address the concerns raised.

---

`Reviewers QZ5c and mi8W` **focused their concern almost exclusively on a single issue**: the comparison against standard imbalance mitigation strategies. In response, we enhanced two strongest baseline models with several widely adopted techniques. In the second round of discussion, we provided the requested statistical significance analysis, conclusively demonstrating that **SymSpectra maintains statistically significant superiority over all enhanced baselines**, further validating our CMI-based method.

Apart from the shared concern, `reviewer mi8W` also questioned the practical relevance of the multi-modal setting and the methodological novelty compared to a baseline model, Alberts et al. In our response, we cited the appropriate prior work to **substantiate the practical utility of our approach** and articulated the fundamental architectural distinctions between our framework and Alberts et al. Additionally, `Reviewer QZ5c` raised several inquiries regarding presentation clarity. We have provided detailed clarifications for these points and **incorporated all corresponding corrections into the revised manuscript**.

---

`Reviewer Rok3` suggested including recent generative models as baselines and sought clarification on the alignment between our theoretical formulation and its implementation. In response, we clarified the **task-specific distinctions** between our framework and the suggested approaches, and incorporated these important works into the revised version. Furthermore, we provided a detailed breakdown of SymSpectra to elucidate its implementation mechanics and structural rationale. Through a direct comparison with the alternative modules suggested by the reviewer, we further highlighted the **specific advantages of our design in effectively tackling the class-imbalanced problem**.

---

Given recent restrictions on reviewer engagement, we respectfully invite the AC to evaluate our comprehensive rebuttal. SymSpectra addresses **the important and under-explored problem of class-imbalanced molecular structure elucidation** through an innovative information-theoretic framework. We believe our rigorous supplementary experiments and clear theoretical proofs **have systematically resolved initial reservations**, specifically regarding baseline comparisons and methodological grounding, thereby strengthening the work's contribution. We sincerely thank the AC for their time and guidance.



Best regards,

Authors

---

### Note · Authors · 2026-01-29

I have read and agree with the venue's withdrawal policy on behalf of myself and my co-authors.

---

### Meta-Review · Area_Chair_vGUD · 2025-12-22

**Summary:**

- Missing comparisons against standard class-imbalance techniques (like under/oversampling, inverse class-frequency loss weighting, focal loss, etc.)
- Limited theoretical justification for SCIB design.
- Unclear methodological novelty relative to SotA methods.
- Strong assumptions (such as Gaussian distributions of spectral representations).
- Unclear practical relevance: Strong baselines are missing.

**Reviewer Concerns:**

In my opinion, the only concern that could be addressed in a fully convincing way is the one about missing comparisons against standard class-imbalance techniques. I think, that all the more fundamental questions about conceptual novelty / limited theoretical justification / unclear validity of assumptions / unclear practical relevance are still open after the rebuttal.

**Reviewer Scores:**

Reviewer mi8W (who asked for comparison experiments and statistical significance analysis) might have raised their score.
I'd not expect significant changes for the other reviewers.

---

### Decision · Program_Chairs · 2026-01-26

Reject